



# Further investigations into the accuracy of infrared-radiofluorescence (IR-RF) and its inter-comparison with infrared photoluminescence (IRPL) dating

Mariana Sontag-González[1], Madhav K. Murari[2], Mayank Jain[3], Marine Frouin[4], Markus Fuchs[1]

[1]Department of Geography, Justus Liebig University Giessen, 35390 Giessen, Germany
[2]Inter University Accelerator Centre, 110067, New Delhi, Delhi, India
[3]Luminescence Physics and Technologies, Department of Physics, Technical University of Denmark Risø Campus, Roskilde, Denmark
[4]Department of Geosciences, Stony Brook University, 255 Earth and Space Sciences Building, Stony Brook, NY 11794-2100, USA

*Correspondence to*: Mariana Sontag-González (Mariana.Sontag-Gonzalez@geogr.uni-giessen.de)

**Abstract.** Infrared radiofluorescence (IR-RF) is an alternative dating technique for potassium feldspar grains, offering a higher signal stability and based on a simpler underlying mechanism than more common luminescence dating approaches. However, its accuracy when tested on known-age samples has so far shown inconsistent results. In this study, we present a refined accuracy assessment using samples that have previously produced unreliable IR-RF ages. Our approach incorporates two major methodological advancements developed over the past decade: elevated temperature measurements using the IR-RF$_{70}$ protocol and sensitivity change correction by vertical sliding. To expand the dose range comparison, we included two additional samples: one expected to be in saturation and another of modern age. Additionally, we evaluated the effect of using a narrower bandpass filter to exclude any signal contributions from potentially contaminating shorter wavelength emissions. Our results following the IR-RF$_{70}$ protocol with sensitivity corrections show an improvement over the original room temperature results. For four out of the seven tested known-age samples spanning 20–130 ka, we obtained results in keeping with the expected doses. Two additional modern samples, however, yielded slight age underestimations. Introduction of a multiple-aliquot regenerative dose (MAR) protocol improved the accuracy of two out of three samples with large sensitivity changes. Finally, we also compared the new IR-RF equivalent doses ($D_e$) to those obtained with the newer trap-specific dating method infrared-photoluminescence (IRPL) for the same samples, including previously published values and new measurements. We observe that with the new improvements the success rate of IR-RF is comparable to that of IRPL.



## 1 Introduction

Infrared radiofluorescence (IR-RF) dating of potassium (K)-feldspar is a technique used to determine the time since sediment deposition (e.g., Trautmann et al., 1998, 1999; Erfurt and Krbetschek, 2003). The method consists of filling available electron traps within the mineral's crystal lattice through continuous ionizing irradiation and quantifying the intensity of the resulting IR emission at ~880 nm (NB this emission was previously reported at 865 nm, see discussion in Sontag-González et al. (2022)). By comparing IR-RF curves obtained after natural burial with those obtained after a known laboratory dose

(e.g., starting at zero dose following complete signal bleaching), the equivalent dose ($D_e$) accumulated since burial can be determined. Dividing the $D_e$ by the sample's environmental dose rate yields the time since burial. The main advantages of IR-RF dating over the more common infrared stimulated luminescence (IRSL) of K-feldspar include a lower anomalous fading rate (Krbetschek et al., 2000) and the large number of data points used to build the dose response curve (DRC).

Buylaert et al. (2012b) determined IR-RF ages for sixteen coarse-grained K-feldspar samples and reported poor

agreement with the independent age controls, which ranged from modern to ~130 ka. For three modern samples, the IR-RF $D_e$ values were shown to be highly sensitive to the bleaching method used between natural and regenerative measurements, with the most accurate results obtained using a solar simulator bleach of at least 4 hours. The IR-RF ages of five young samples (~100–150 Gy) were overestimated by ~20–70%, while those of eight older samples (~200–300 Gy) were underestimated by ~20–40%. Over the past decade, significant methodological improvements in IR-RF dating have been

developed, including measurement at 70°C instead of at room temperature (Frouin et al., 2017) and a vertical slide correction for sensitivity changes occurring between the two IR-RF measurements, initially proposed by Buylaert et al. (2012b) and implemented by Murari et al. (2018). When introducing the IR-RF$_{70}$ protocol, Frouin et al. (2017) presented results for six samples, five of which matched the quartz-based ages (at 2σ). These new developments warrant a re-analysis of the previous findings on the accuracy of IR-RF ages reported by Buylaert et al. (2012b).

In parallel, Prasad et al. (2017) proposed a new dating method for K-feldspar based on IR photoluminescence (IRPL), which is relevant in the context of IR-RF dating. Both methods involve the emission stemming from electrons relaxing from the excited to the ground state of the K-feldspar principal trap. In IR-RF, these electrons originate from the valence band and get trapped during the process, whereas, in IRPL, the electrons are already trapped but are stimulated to the excited state with IR stimulation. Importantly, in IRPL, the electrons do not leave the trap, leading to a steady-state signal

(i.e., electrons continuously 'bouncing' between the excited and ground states of the principal trap during IR stimulation) and thereby representing a non-destructive measurement technique. Kumar et al. (2018) identified two IRPL emissions resulting from stimulation with an 830 nm laser, which are centred at 880 and 955 nm (IRPL$_{880}$ and IRPL$_{955}$, respectively). They also demonstrated that these two IRPL emissions correspond to the two known IR-RF emissions and suggested that they arise from the same defect (i.e., the principal trap) but with slightly different environments (i.e., neighbouring atoms or

functional groups). Using a single-aliquot regenerative (SAR) protocol, Kumar et al. (2021) dated eleven sediment samples with IRPL, nine of which had also been dated using IR-RF by Buylaert et al. (2012b). They observed good agreement



between the IRPL ages and the independent age controls for eight samples ranging ~100–300 Gy, but obtained overestimated ages for three modern or very young samples. If the IRPL and IR-RF signals indeed originate from the same trap reservoir, they would be expected to yield similar $D_e$ values, assuming there is no contamination from other emissions as

well as negligible sensitivity changes during the measurement process. Kumar et al. (2021) observed poor agreement between their IRPL ages and IR-RF ages of the same samples reported by Buylaert et al. (2012b), which were obtained using room temperature IR-RF measurements, and without a correction for sensitivity changes. A comparison between IRPL measurements and the new improved IR-RF protocol (Frouin et al., 2017; Murari et al., 2018) are yet to be made.

In addition, the upper dating limit of both IR-RF and IRPL are still poorly understood. In the case of IR-RF,

previous studies suggest the onset of saturation of the natural signal at around 1000 Gy. This value is based on measurements of a Triassic age sample expected to be in saturation (Murari et al., 2021b), measurements of a sequence of known-age samples from the Chinese Loess Plateau (Buchanan et al., 2022), and a study of long-term signal stability using artificially-added gamma doses (Kreutzer et al., 2022a). These findings are at odds with the DRCs obtained in the laboratory, which continue to grow over >3000 Gy. The upper dating limit of IRPL has not been studied using natural samples yet. However,

the characteristic saturation dose (85% saturation) of laboratory-irradiated DRCs ranges between ca. 1400 Gy and 2500 Gy (Kumar et al., 2021) depending on the protocol used (e.g., with or without preheat). Comparison of the IRPL and IR-RF field-saturation doses will help determine whether the relatively low upper limit of ~1000 Gy for IR-RF dating is a characteristic of the principal trap (e.g., due to signal instability) or whether it is caused by IR-RF specific issues. One potential issue is an overlap of the targeted IR-RF emission (centred at 880 nm) with a neighbouring red emission, which has

been reported to be thermally unstable (Krbetschek et al., 2000). This overlap can be problematic for $D_e$ estimation by making the natural dose curve not directly comparable to the regenerative dose curve, as the red signal would have already decayed in the natural sample but still contribute to the total signal in the regenerated dose curve. Spectroscopic measurements of a field-saturated sample by Sontag-González and Fuchs (2022) reported differences in mean $D_e$ of ~400 Gy between integration of the dose ranges 810–850 nm and 850–890 nm, confirming that the detection window needs careful

consideration. Shifting the detection window further into the IR would reduce the contribution from the red emission, potentially leading to more accurate $D_e$ values.

Here, we re-analyse the original data for the sixteen samples dated by Buylaert et al. (2012b) to include a sensitivity change correction (vertical slide) and present ten new IR-RF $D_e$ obtained using the IR-RF$_{70}$ protocol for samples with independent age controls. The IR-RF$_{70}$ dataset includes eight samples originally used by Buylaert et al. (2012b), one

additional modern sample and one sample expected to be in field-saturation. We also test a different detection window to assess whether it improves the accuracy of IR-RF$_{70}$ $D_e$ values. In addition to the previous SAR measurements, we also test the suitability of an IR-RF multiple-aliquot regenerative dose (MAR) protocol to overcome large sensitivity changes. Finally, we compare the IR-RF$_{70}$ results with IRPL ages for ten samples. IRPL data of seven of these samples have been published by Kumar et al. (2021). Additionally, we measure three new samples following a multiple elevated temperature

(MET) post-IR IRSL (pIRIR) protocol.



## 2 Samples and methods

### 2.1 Sample selection and preparation

Eighteen samples (Table 1) were selected based on their independent age controls. Seventeen of these were prepared in the luminescence laboratory at Risø DTU following standard procedures. After initial wet-sieving, grain size fractions of 90–150, 90–180, 150–212, 180–250 or 150–250 µm in diameter were treated with 10% hydrogen chloride (HCl) and 10% hydrogen peroxide ($H_2O_2$) solutions to remove carbonates and organic matter, respectively. K-feldspar grains were then isolated using a heavy liquid solution with a density of 2.58 $g/cm^3$ and etched with hydrogen fluoride (HF; 10%) for 40 min to remove the alpha-irradiated outer layer. A final HCl treatment dissolved any contaminating fluorides. One additional sample, Gi326, was prepared in the luminescence laboratory in Giessen, using the same chemical treatments, except for HF-etching. This sample is from a Triassic sandstone and has been used previously to test the field saturation of IR-RF$_{70}$ (Murari et al., 2021b).

### 2.2 Instrumental setup for IR-RF measurements

The original IR-RF luminescence measurements from Buylaert et al. (2012b) conducted at room temperature (IR-RF$_{RT}$), were obtained using a Risø TL/OSL DA-20 reader (Bøtter-Jensen et al., 2010) with a specific attachment (Lapp et al., 2012), where the signal is transmitted through an optical light-guide and filtered with a Chroma D900/100 interference filter before reaching a Hamamatsu H7421-50 photomultiplier tube (PMT). The effective detection window of this filter and PMT combination has a full width at half maximum (FWHM) spanning ~850–875 nm. Large aliquots, containing thousands of grains mounted on stainless steel discs with silicone oil, were measured according to the IR-RF$_{RT}$ protocol outlined in Table 2. Aliquots were bleached between the natural and regenerative dose steps using either built-in UV LEDs (395 nm) or an external SOL2 solar simulator (Hönle AG).

New radiofluorescence measurements were performed at 70°C (IR-RF$_{70}$) using a *lexsyg research* device (Freiberg Instruments GmbH; Richter et al., 2013) in Giessen (Germany) equipped with an annular beta source ($^{90}Sr/^{90}Y$) calibrated with a standard quartz sample (Risø calibration quartz batch 200). We assume an uncertainty of 5% for this calibration, which is added in quadrature to the mean $D_e$ standard error of each sample. Measurements in section 3.6 were all conducted at the University of Oxford (UK) on the same type of device; further details can be found in Sontag-González et al. (2024b). The radiofluorescence was filtered through a Chroma D850/40 interference filter before being detected by a Hamamatsu H7421-50 PMT with an effective detection FWHM spanning ~825–860 nm. Additional experiments in section 3.4 tested a different interference filter centred at 880 nm with an FWHM spanning ~875–885 nm (Quantum Design 880HC10). Medium-sized aliquots, containing hundreds of coarse grains, were mounted on stainless steel cups with silicone oil. For the 'single-grain' measurements in section 3.6, an individual grain was manually placed on each sample holder and fixed with silicone oil. The new measurements followed the IR-RF$_{70}$ protocol outlined in Table 2. Aliquots were bleached using a built-in LED-based solar simulator, according to the relative intensities in Frouin et al. (2015) and using the following power





output: 365 nm (9 mW), 462 nm (55 mW), 525 nm (47 mW), 590 nm (32 mW), 625 nm (100 mW), and 850 nm (84 mW). In a separate bleaching test, some aliquots were bleached by direct sunlight exposure (see section 3.5).

## 2.3 IR-RF data analysis

Data analysis of both the previous and the new IR-RF measurements was conducted in an R programming environment v4.0.2 (R Core Team, 2020). $D_e$ values were obtained for each aliquot by sliding the 'natural dose' curves onto the 'regenerative dose' curves using either only a horizontal slide or both horizontal and vertical slides utilizing the function 'analyse_IRSAR.RF' from the 'Luminescence' (v0.9.19) package (Kreutzer et al., 2012, 2022b). The IR-RF$_{70}$ curves from a representative aliquot are shown in Fig. 1, where the extent of vertical and horizontal sliding of the natural dose curve is shown as grey bars.

Following conventional procedure, the initial channels of each natural measurement were ignored during the analysis to account for the 'initial rise' effect, i.e., an unexpected signal increase at the beginning of natural and regenerative dose curves whose origin is still not well understood but which affects the initial ~20 Gy (e.g., Buylaert et al., 2012b; Murari et al., 2021a). The initial rise phenomenon is apparent in the first few data points of the curves shown in Fig. 1; in this case, removing the first channel of the natural dose curve would be sufficient to make the natural and regenerative dose curves comparable over the dose range of interest. The effect of rejecting the initial channels will be discussed in section 3.3.

## 2.4 IRPL equipment and procedure

We measured the IRPL emissions centred at 880 and 955 nm for three samples using the same equipment as Kumar et al. (2021). We used the IRPL attachment (Kook et al., 2018) to the Risø TL/OSL reader with two photomultiplier tubes: a Hamamatsu H10330C-25 in combination with a bandpass interference filter centred at 950 (FWHM of 50 nm) for the IRPL$_{955}$ emission and a Hamamatsu H7421-50 in combination with either a bandpass interference filter centred at 880 nm (FWHM of 10 nm) for the IRPL$_{880}$ or BG39 and BG3 filters for the IRSL emission. IR LEDs (850 nm with a power density of ~250 mW/cm$^2$ at sample position) were used as the stimulation for IRSL. The IRPL stimulation consisted of an external laser light source (830 nm with a power density of ~3 mW/cm$^2$ at sample position) in a pulsed excitation mode (50 µs on-time followed by 50 µs off-time) for 5 s. IRPL was only measured during the off-time, rejecting the first 3 µs to avoid contamination by the excitation source (PMT gate 53–100 µm).

The measurement protocol consists of an MET pIRIR protocol based on Protocol A of Kumar et al. (2021) and using IR stimulation temperatures of 50°C, 90°C, 130°C and 290°C (IRSL$_{50}$, pIRIR$_{90}$, pIRIR$_{130}$, pIRIR$_{290}$, respectively) with room temperature IRPL measurements interspersed between each step (Table S1). The preheat step was set to 320°C (60 s) and the IR stimulation at 290°C (100 s) was used to remove residual signal prior to each irradiation step (cleanout). IRPL signals for each emission were calculated as the integral of the 5 s long measurements, subtracting as background the residual IRPL obtained after the 290°C IR cleanout (step 16 or 17 in Table S1 for IRPL$_{880}$ or IRPL$_{955}$, respectively). Fig. S1 and Fig. S2 show examples of the IRPL curves used to obtain the signal and background, respectively. Fig. S3 displays





examples of the resulting DRC. IRSL signals were also determined for comparison, consisting of the integral of the first 2 s with the last 10 s used for background subtraction. Sensitivity change corrections for IRSL and IRPL consisted of dividing the signal obtained from the natural or various doses ($L_n$ or $L_x$, respectively) by that obtained from a test dose ($T_n$ or $T_x$) for each SAR cycle. The IRPL and IRSL DRCs were fitted with double exponential functions. Aliquots were only considered for $D_e$ estimation if they passed the following criteria: minimum signal brightness ($T_n$ more than three sigma of the

background level and a relative standard error of $T_n$ <10%), reproducibility (recycling ratio within 10% of unity) and low signal carry-over (recuperation <15%).

## 3 IR-RF $D_e$ estimation

### 3.1 Effect of vertical slide correction

The original mean $D_e$ values obtained by Buylaert et al. (2012b) using an IR-RF$_{RT}$ protocol are reproduced in Fig. 2a (open

circles). When using the internal UV LEDs for the bleaching steps between the natural and regenerative dose steps, the $D_e$ values are either over- or underestimated (at 2σ) for all sixteen samples. By replacing the UV bleaching step with an external bleach in a solar simulator (SOL2) for a subset of six samples, Buylaert et al. (2012b) observed an improvement in results, but the $D_e$ values were still under- or overestimated for five of the six samples tested (Fig. 2a, green open circles).

Application of the new vertical slide correction for sensitivity changes led to an increase in $D_e$ values for all

samples except for one of the modern ones (Table 3 and Fig. 2a, filled circles). This adjustment resulted in eight samples aligning with the expected $D_e$ (at 2σ), but it also increased the overestimation for most of the other samples. A similar pattern was observed for the aliquots bleached with UV LEDs and with SOL2. For the UV LED dataset, the sensitivity change correction led to an increase in the scatter of the $D_e$ distribution of most samples, evident from the increased standard errors of the mean (Fig. 2a). These results suggest that sensitivity change corrections alone do not improve the accuracy of IR-RF$_{RT}$

results.

### 3.2 Effect of measurement temperature

Of the sixteen samples analysed in section 3.1, we selected eight for new measurements at 70°C, keeping only 1–2 samples per site. The new measurements using the IR-RF$_{70}$ protocol showed a better agreement with the expected values (Fig. 2b). Four of the samples have $D_e$ values matching expectations at 1σ. Interestingly, for these four samples, results with and

without the sensitivity change correction were indistinguishable from each other at 1σ. For three additional samples, we noted a large overestimation and, for a modern sample, we noted an underestimation regardless of whether the sensitivity change correction was applied or not.



### 3.3 Sliding range of IR-RF$_{70}$ natural dose curve

For the analyses shown in Fig. 2a–b, we used the same dose ranges in the sliding procedure as in the original analysis of
Buylaert et al. (2012b) to allow for a more direct comparison. In their study, between 1 Gy and 33 Gy were removed from
the beginning of each natural dose curve to avoid channels affected by the initial rise, which hinder the sliding algorithm.
This approach required an aliquot-specific manual adjustment. For all aliquots, we measured the natural dose curves over a
relatively long dose range of ~1800 Gy when using the IR-RF$_{70}$ protocol. However, for the results shown in Fig. 2a–b we
also adjusted the total lengths of the natural dose curve segments used for sliding to match those of Buylaert et al. (2012b),
which spanned 5–475 Gy, depending on the sample. Now, we will investigate whether the choice of the natural dose curve
segment used for sliding affects the resulting D$_e$ value with the IR-RF$_{70}$ protocol. For the following, we added two new
samples to the dataset, which were not analysed by Buylaert et al. (2012b): a modern sample (B12) and a field-saturated
sample (Gi326). At least three aliquots were measured per sample.

First, we assessed the dose range that should be removed from the beginning of the natural dose curves to mitigate
the initial rise effect. We compared the mean D$_e$ values resulting from using the initial ~150 Gy of the natural dose curve,
while rejecting various initial amounts: 0 (no rejection), 2, 5, 11, 17, 36 or 72 Gy for each of ten samples (Fig. 3a–b). Note
that the rejected channels are still considered for D$_e$ calculation as laboratory-added dose even though they are excluded from
the sliding algorithm to assess goodness-of-fit. Using only a horizontal slide analysis (Fig. 3a), we observed no significant
change in D$_e$ except for the modern samples, for which only the iterations without any rejection matched the expected doses.
When applying the sensitivity change correction (Fig. 3b), we observed a pattern of gradual decrease in D$_e$ values of up to
17% (relative to the no rejection scenario) for all samples.

We then repeated this test using a longer segment of the natural dose curve (~600 Gy) while rejecting the initial 0
(no rejection), 2, 5, 36, 108 or 301 Gy for each of the ten samples (Fig. 3c–d). For the modern samples, only the iterations
with no rejection matched the expected dose (at either 1 or 2σ) and increasing rejection led to a gradual decrease of the mean
D$_e$. For the other samples, minimal change was observed when rejecting less than 35 Gy in either analysis type. However,
rejecting a larger initial dose range resulted in a gradual increase in D$_e$ for most samples. Comparing results using shorter
(~150 Gy) and longer (~600 Gy) natural dose curve segments with sensitivity change correction (Fig. 3b and 3d,
respectively) reveals that the length of the segment has a higher impact on D$_e$ than the amount of rejected initial channels. If
the natural dose curve is sufficiently long, up to ~35 Gy can be removed to address the initial rise effect.

However, the question remains as to how long the natural dose curve should be. We tested this by fixing the
beginning of the curve (rejecting the initial 2 Gy to avoid the channels most affected by the initial rise) and changing the
length of the curve to span ~150, 300, 600, 900, 1200 or 1800 Gy (Fig. 3e–f). For the two modern samples, we observed
minimal D$_e$ variation. For the other samples and without a sensitivity change correction, we observed a gradual increase in
D$_e$ with increasing length of the natural dose curve (Fig. 3e). Relative to the shortest segment (~150 Gy), the D$_e$ values
increase by between 2% and 25% with the larger increases (11–23%) observed for the field-saturated sample and for the





three samples with poor matches to the expected doses. Note that for the field-saturated sample, the longest segment yielded only a minimum dose estimate due to the limit of the regenerative dose curve (i.e., the analysis algorithm restricts sliding the natural dose curve further than the end of the regenerative dose curve). The remaining four samples showed only 2–5% increase. We observed a similar pattern when applying the sensitivity change correction, but the trend of increase in $D_e$

generally begins only for segments ≥600 Gy, suggesting the natural dose curve should not be longer than that. For the field-saturated sample, $D_e$ values increased up to 26%. For the other non-modern samples, the highest $D_e$ values were 5–25% higher than the lowest ones, depending on the sample.

The results shown in Fig. 3e–f suggest sensitivity changes occur during measurement of the natural dose curve that are not corrected by vertical sliding, as this method only addresses changes between the end of the natural and beginning of

the regenerative dose curves. A comparison of the fit quality between the natural and regenerative dose curves for samples with small and large sensitivity changes is shown in Fig. S4.

To further investigate sensitivity changes, we repeated the $D_e$ estimations for all samples using segments of the natural dose curves spanning ~300 Gy, but beginning at either 2, 301, 602, 903, 1205 or 1506 Gy (Fig. 3g–h). We observed a gradual change in $D_e$ for all samples with both analysis types. Without a sensitivity change correction, $D_e$ values increased

by up to 22–294% (relative to the initial segment) for the non-modern samples. The three samples whose $D_e$ values consistently overestimate (sample No. 092202, 075406 and 072255) had much larger increases (78–294%) than the four other non-modern samples (22–36%). The modern samples had either an increase or a decrease of 60–75 Gy. For the field-saturated sample, the final three segments yielded only a minimum dose estimate due to the limit of the regenerative dose curve. We observed a $D_e$ increase of up to 32% for the field-saturated sample but expect that that number would be higher if

the regenerative dose curve had been longer and thus allowed for an absolute value for the final three segments. With the sensitivity change correction, the $D_e$ increases were less pronounced: up to 49–116% for the three 'problematic' samples as well as the field-saturated sample and up to 12–28% for the other non-modern samples.

Overall, the length of the natural dose curve segment used for sliding affects the resulting $D_e$ values for most of the tested samples. For short segments (e.g., ≤151 Gy), results using the vertical slide analysis are very sensitive to an initial

channel rejection. Excessively long segments might lead to a dose overestimation possibly due to sensitivity changes occurring during the natural dose curve measurement. A segment of ~600 Gy seems to be relatively insensitive to the rejection of the initial channels for non-modern samples. Mean $D_e$ values for all samples using this segment are given in Table 4 and shown in Fig. 2c.

## 3.4 Detection window

The observed dose-dependent mismatch between the natural and the regenerative dose curves could be due to varying contributions from neighbouring emissions to the total signal. To test whether the observed $D_e$ variations are caused by a contribution of the red RF emission (centred at ca. 700–730 nm; Trautmann et al., 1998; Krbetschek et al., 2000), we shifted the detection window further into the IR by replacing the bandpass filter centred at 850 nm (FWHM: 40 nm) with one





centred at 880 (FWHM: 10 nm) and repeating all measurements and analyses discussed in section 3.3. We chose a filter with
a narrower detection window to avoid emissions in shorter or longer wavelengths than the desired 880 nm RF emission. If
the red emission contributed to the total signal, we would expect a higher $D_e$ accuracy and lower $D_e$ variations with segment
choice when using the new filter. Fig. S6 shows the resulting mean $D_e$ of three aliquots per sample.

        We generally observed similar patterns with both filters, though measurements with the 880 nm filter showed
slightly more variability, probably due to the lower signal intensity with the narrower filter (compare Fig. S4 and S5). Using
a short segment of ~150 Gy and rejecting up to the initial 70 Gy with the vertical and horizontal slide analysis (Fig. S6b), we
observed a gradual change in mean $D_e$ for most samples: an increase for the field-saturated sample and a decrease for the
others. The shortest segment (71–147 Gy) yielded relatively high standard errors of the mean, suggesting it is too short for
reliable vertical sliding.

        Using a longer natural dose curve segment (~600 Gy) and rejecting up to the initial ~300 Gy (Fig. S6c–d), we
observed a slight increase in mean $D_e$ values for some non-modern samples if ≥35 Gy are rejected. For the modern samples,
rejecting the initial channels leads to a gradual increase in underestimation for both samples and analysis types.

When keeping the beginning of the natural dose curve segment fixed at 2 Gy and extending its end to 147, 294, 588, 881,
1175 or 1763 Gy (Fig. S6e–f), we observed no significant change in mean $D_e$ for the two modern samples. For the other
samples, we observed the same pattern as for the first filter: without a sensitivity change correction, the $D_e$ increases by 2–
20% depending on the sample whereas, with a sensitivity change correction, there is an initial decrease for some samples and
for most samples an increase starting from ~300 Gy. The highest $D_e$ values were 6–25% higher than the lowest ones,
depending on the sample.

        When using a curve segment spanning ~300 Gy, but starting at different doses (2, 293, 587, 881, 1175 or 1469 Gy),
we still observed a large gradual $D_e$ increase for most samples (Fig. S6g–h). Without a sensitivity change correction, the
modern samples' mean $D_e$ varied by 20–30 Gy, whereas the non-modern samples' mean $D_e$ increased by 10–124% relative
to the initial segment, depending on the sample (Fig. S6g). Overall, the maximum increases were lower than with the
previous filter (22–294%) but more samples displayed a large variation (six vs. the previous four). Thus, shifting the
detection window further into the IR did not necessarily decrease the mismatch between the natural and regenerative dose
curves. With the sensitivity change correction, $D_e$ changes were less consistent (Fig. S6h): there were increases of up to
211% relative to the initial segment, but the $D_e$ progression across the samples was less monotonic than with the previous
filter. This is probably caused by higher uncertainty in the sliding algorithm due to the lower signal intensity.

        The mean IR-RF$_{70}$ $D_e$ values for all samples using the initial ~600 Gy segments obtained with the 880 nm filter
(FWHM: 10 nm) are given in Table 4 and compared with those using the broader filter (850 nm; FWHM of 40 nm) in Fig.
2c (for sample Gi326, see Fig. 6b–c). Overall, the differences between both filters are small. The new, narrower filter
resulted in slightly lower mean $D_e$ values for seven out of ten samples, but the mean $D_e$ values are statistically
indistinguishable (at 2σ) for all samples.



### 3.5 Bleaching test

Given the large differences in IR-RF $D_e$ observed by Buylaert et al. (2012b) when using the internal UV LEDs or an external solar simulator (SOL2) for the signal bleach between the natural and regenerative dose steps, we tested the efficacy of the

internal solar simulator for IR-RF$_{70}$ measurements by comparing it to a solar bleach. We used three samples with poor matches to their expected doses (sample No. 092202, 075406 and 072255) and one sample that matched its expected dose (H22553). Three aliquots per sample received an external solar bleach between the natural and regenerative steps by placing them on a windowsill for one week in late May in Giessen, Germany. The aliquots were exposed to cumulative ~82 h of daylight, including at least 8 h of direct sunlight. $D_e$ values obtained with vertical slide correction were statistically

indistinguishable (at $1\sigma$) from those obtained using the internal solar simulator (Fig. 2d), suggesting that the internal solar simulator bleach of 24 817 s (~7 h) used here is sufficient to reduce the IR-RF$_{70}$ signal to a level comparable with natural sunlight conditions. The observed $D_e$ overestimation for three of these samples is, thus, unlikely due to bleaching issues during the IR-RF$_{70}$ protocol.

### 3.6 Single-grain variability and the initial signal rise

Previous research has suggested differences in IR-RF characteristics at the single-grain level which would be averaged in the multi-grain aliquots used in the present work (Trautmann et al., 2000; Frouin et al., 2017; Mittelstraß and Kreutzer, 2021). For example, our earlier work on sample H22550 (Sontag-González et al., 2024b) revealed that only three out of six measured grains produced the expected IR-RF signal (i.e., decreasing with dose, referred to as Type #1). One additional grain exhibited an initial signal rise followed by signal saturation at ~100 Gy (termed Type #2). The remaining two grains

showed a flat signal shape indistinguishable from that of an empty sample holder. These results suggest potential issues for multi-grain measurements. By mathematically adding the IR-RF signal (i.e., a synthetic aliquot) obtained from individual grains of sample H22550 by Sontag-González et al. (2024), we modelled the signal from multi-grain aliquots with different proportions of the two grain types (Fig. 4). If at least one third of the grains in a multi-grain aliquot yield the expected IR-RF signal, they will dominate the total signal and, overall, the signal shape will follow the Type #1 grain pattern (Fig. 4, inset).

Interestingly, our modelled signal displays an 'initial signal rise' which decreases with a decrease of the proportion of unwanted Type #2 grains. This suggests that (i) the 'initial signal rise' originates from signal contamination by presumably non-K-feldspar minerals and (ii) the signal shape at doses higher than ~100 Gy is not significantly compromised by this phenomenon.

### 4 An IR-RF MAR protocol

As indicated in section 3.3, some samples appear to suffer from significant progressive sensitivity changes during the first laboratory irradiation, thereby making the natural and regenerative dose curves not directly comparable. More importantly, the vertical slide is not an effective technique to correct for such progressive sensitivity changes, as it only corrects for



changes in signal intensity but still assumes that the curve shapes are comparable. In luminescence dating, MAR procedures (Aitken, 1998) can be implemented to circumvent sensitivity changes when the typical test dose correction in a SAR protocol is deemed insufficient for either quartz (e.g., Lu et al., 2007; Ankjærgaard, 2019) or K-feldspar applications (e.g., Li et al., 2013). In MAR protocols, the natural and regenerative data sets stem from different aliquots, with the latter being bleached to remove the natural signal prior to a laboratory irradiation.

We tested the suitability of an IR-RF MAR protocol to overcome the sensitivity changes induced by the first laboratory irradiation of an aliquot. The protocol consisted of obtaining the natural dose curves for several aliquots following the same steps as in a SAR protocol (for simplification, the same dataset was used as in section 3.3), whereas the regenerative dose curves (one per sample) were obtained from new aliquots in which the natural dose curve step and associated preheat were skipped (MAR protocol in Table 2). To account for inter-aliquot differences in mass and signal intensity, each SAR natural dose curve was scaled by the ratio between the first data point of the MAR and SAR regenerative dose curves, as exemplified in Fig. S7. An instrumental background was subtracted from each curve prior to normalisation, using the mean signal intensity of a 1000 s long measurement of an empty sample holder (12 961 cts/channel).

We repeated the MAR $D_e$ analysis using varying natural dose curve segments, as previously described for the SAR protocol in section 3.3. As shown in Fig. S8, use of different segments led to different $D_e$ values for all ten investigated samples, though some differences in pattern are of note. Unlike the SAR analysis, in which use of an increasingly long natural dose curve segment led to higher $D_e$ values for all samples (Fig. 3f), with the MAR analysis, increasingly long segments (i.e., 2−151 Gy to 2−1808 Gy) led to a better agreement with expected $D_e$ values for five non-modern samples and very small changes for the remaining four samples of known age (Fig. S8d). For this reason, we also considered how many channels should be removed from the beginning of the 1808 Gy-long natural dose curve (Fig. S8c). We found that rejecting the initial 108 Gy was the best compromise to achieve a good agreement with expected $D_e$ values across the non-modern samples (Fig. 5a and Table 4). The MAR protocol (natural dose curve segment of 108−1808 Gy with vertical and horizontal slide) yielded $D_e$ in agreement with the expected values (at 2σ) for five out of the seven non-modern samples of known age, including for two of the three samples identified as 'problematic' in section 3.3.

## 5 Comparison of IR-RF and IRPL

A subset of the samples used in section 3 to evaluate different IR-RF protocols have also been used to assess the accuracy of IRPL (Kumar et al., 2021), allowing us to directly compare both methods. To create a more comprehensive dataset, we measured three additional samples (075406, A8 and Gi326) following an IRPL protocol and using the same equipment as Kumar et al. (2021). For each sample, between three and seven aliquots were measured.

Since the IRPL signal measurement is mostly non-destructive, it can be taken at several steps in the protocol, yielding several IRPL $D_e$ values for each emission (i.e., both before and after the preheat as well as before each IRSL step).





Note that the IRPL steps after the IRSL$_{290}$ step are merely used to determine a background signal (see Fig. S2) and do not yield a D$_e$ value. In this study, we measured the 880 nm and the 955 nm IRPL emissions. The sequential IRSL steps can also each be used to obtain a D$_e$ value, thus resulting in a total of fourteen D$_e$ values for each aliquot (see Table S1).

The mean D$_e$ value for each IRPL signal of the field-saturated sample Gi326 is shown in Fig. 6a. With the exception of the IRSL$_{50}$ D$_e$, which is expected to be significantly affected by fading, all other D$_e$ values are similar, ranging from ~1100 to

~1400 Gy. After the IRSL$_{50}$ step, we observed a convergence of the different IRPL signals to ~1200 Gy. This value is slightly lower than the pIRIR$_{290}$ mean D$_e$ value at 1293 ± 14 Gy (Fig. 6a) and the IR-RF mean sensitivity change corrected D$_e$ values, which are also at ~1300 Gy (Fig. 6b–d). We do not notice a significant change between the pIR$_{50}$IRPL and the following IRPL mean D$_e$ values, suggesting there is no sequential removal of unstable IRPL signal after increasingly hot IRSL steps beyond 50°C for this sample. A similar behaviour is observed for the three pIRIR mean D$_e$ values, though the

large uncertainties in the pIRIR$_{90}$ and pIRIR$_{130}$ mean D$_e$ values might obscure the expected step-wise increase in signal stability with increasing IRSL temperature (e.g., Li et al., 2013).

Kumar et al. (2021) stated that the IRPL$_{880}$ after the preheat step (APh-IRPL$_{880}$) and the pIR$_{50}$IRPL$_{955}$ signals are the most promising ones for sediment dating, based on a better agreement with the expected doses. Mean D$_e$ values for these two signals, including two new samples are shown in Fig. 5b together with the most promising IR-RF signal. The values for

other IRPL signals are shown in Fig. S9 (new samples) and Fig. S10 (all samples). The mean D$_e$ values from the pIR$_{50}$IRPL$_{955}$ signal in Fig. 5b are in good agreement (at 2σ) with the expected doses for all non-modern samples and for six out of seven samples using the APh-IRPL$_{880}$ signal including two samples strongly overestimated with IR-RF$_{70}$. However, IR-RF$_{70}$ is in much better agreement than any of the IRPL signals for the two modern samples.

## 6 DRC shape and signal saturation

The results presented in section 3 showed variability in the IR-RF$_{70}$ signal across samples regarding their potential to yield the expected dose and the purported sensitivity changes occurring during the natural dose curve measurement. To assess the variability of the IR-RF curve shapes, we plotted all curves obtained with the 850 nm bandpass filter (FWHM: 40 nm) in Fig. 7. In the natural dose curves (Fig. 7a), the modern and field-saturated samples are distinguishable by their steeper and flatter shapes, respectively (highlighted in pink and grey, respectively). The regenerative dose curves (Fig. 7b) show sample-

dependent variability in signal brightness and in the 'background' level after ~4000 Gy. However, after subtracting the signal value at 1800 Gy as background and normalising the natural dose curves to the signal value at 3 Gy (to discount patterns caused by the initial signal rise), the natural DRC shapes of all samples are relatively similar (Fig. 7a, inset), with small differences observed for the field-saturated and modern samples as well as for samples 092202 and 075406. In contrast, an equivalent analysis of the regenerative dose curves yielded larger differences in DRC shape (Fig. 7b, inset). All samples

follow a similar pattern, except for samples 092202 and 075406 (highlighted in blue and orange in Fig. 7), which saturate later and earlier than the other eight samples, respectively. These two samples were also identified as problematic in section





3.3 when using different segments of the natural dose curve for sliding. An early-saturating DRC could potentially explain gradually more inaccurate $D_e$ values (as observed in Fig. 3g−h) due to the higher uncertainties when using a flatter curve for sliding (i.e., near saturation). However, the same behaviour of gradually higher overestimation was observed for sample

092202 with a late-saturating curve and for sample 072255, whose curve shape follows the consensus. Thus, variability in saturation behaviour does not seem to be the root cause of the observed $D_e$ inaccuracy. The shapes of the MAR DRCs (in Fig. 7c) follow the same overall pattern across samples as the regenerative dose SAR curves.

We also compared the IR-RF$_{70}$, pIR-IRPL and pIRIR DRC shapes up to ~4000 Gy for representative aliquots of the field-saturated sample Gi326 (Fig. 8). Note that only the pIR$_{50}$IRPL$_{955}$ signal is shown in Fig. 8, but all IRPL signals

measured after the preheat step have very similar DRC shapes (see Fig. S3). Likewise, the IR-RF$_{70}$ DRCs of sample Gi326 measured with an 880 nm (FWHM: 10 nm) or an 850 nm (40nm) bandpass filter are very similar in shape (compare Fig. S4 and Fig. S5). By plotting the regenerative IR-RF curves onto an inverted axis, we could directly compare their shapes to those of the other signals, despite their decrease with increasing dose. To assess only the dynamic range of the IR-RF curve, we accounted for the high background signal typical of IR-RF by subtracting the median signal of the last 100 channels (~60

Gy) from all data points. The curves were then normalised to their maximum signal values. The DRC behaviour is quite similar for all three signals using SAR protocols. None of the three signals seems to provide a significant advantage in terms of onset of signal saturation and thereby an extension of the datable age range. The mean $D_e$ values obtained with IR-RF, IRPL and pIRIR in sections 3, 4 and 5 (summarized in Fig. 6) suggest an upper limit of 1000–1300 Gy for these signals. By vertically projecting the mean $D_e$ of each signal onto the corresponding curve, we can qualitatively assess the percentage of

signal saturation. Assuming the signals do not continue to grow past 4000 Gy, field-saturation was achieved at ~80–85% saturation in all three SAR-based curves; the MAR IR-RF field-saturation was slightly higher at 87−89%. If the curves had been continued until full signal saturation, the percentages would be slightly lower. These results suggest a similar small long-term signal instability for the three emissions in this sample.

## 7 Discussion

In the present work, we re-analysed the IR-RF$_{RT}$ data of sixteen samples previously examined by Buylaert et al. (2012b) using an updated data analysis method for $D_e$ estimation known as vertical slide correction. This technique accounts for sensitivity changes occurring between the natural and regenerative dose curve measurements. Application of this correction method improved the accuracy for half of the samples but led to overestimations for most of the other half (see Fig. 2a and Table 3).

We also re-measured eight of these samples with a newer and modified measurement protocol and included two additional samples of known age to expand the dataset for a following comparison with IRPL. Our new measurements differed from the original ones in three main aspects: the measurement temperature, the length of the natural dose curve measurement and the detection window. Measuring RF at an elevated temperature to keep the shallow traps empty following





the IR-RF$_{70}$ protocol (Frouin et al., 2017) had a large impact on D$_e$ and, hence, on improving the accuracy of the resulting
ages (see Fig. 2b). The temperature of 70°C was suggested by Huot et al. (2015), corresponding to the temperature at which
shallow traps are empty and also the temperature spontaneously reached by the thermocouple during the bleaching step in
their *lexsyg* research device. This temperature was then implemented and tested in Frouin et al. (2017) based on a series of
IR-RF measurements of nine samples. As new devices have been developed and given the temperature sensitivity of K-
feldspar signals, users should conduct systematic IR-RF tests (e.g., dose recovery) to assess the protocol parameters for their
samples.

Determining the optimal length of the natural dose curve used for sliding is challenging. In Buylaert et al. (2012b),
the natural dose curve length varied between 5 and 475 Gy, depending on the sample. If the natural and regenerative dose
curves followed the same decay, we would not expect any difference in D$_e$ values from natural dose curves of different
length, provided the segment is long enough for a statistically reliable slide. We tested this hypothesis by measuring a long
natural dose curve of ~1800 Gy for all aliquots but repeating the D$_e$ analysis using segments of different lengths. Contrary to
our expectations, we observed continuing changes in mean D$_e$ values when varying the natural dose curve length from ~150
to ~1800 Gy (Fig. 3f). It is important that the segment not be too short to ensure a good comparison between the natural and
regenerative dose curves, especially when using the vertical slide. Here, we determined that a length of ~600 Gy for the
natural dose curve yielded the most promising mean D$_e$ values in a SAR protocol (Fig. 2c–d and Table 4), but do not
consider this to be a definitive value until the mechanism behind the changes is explained. A length of ~600 Gy has also
been used in recent IR-RF studies (e.g., Murari et al., 2021b; Buchanan et al., 2022; Kreutzer et al., 2022a). As for the
vertical slide sensitivity change correction, its application led only to a small difference in the mean D$_e$ of the four samples
that agreed with the expected doses (at 2σ) and the two modern samples. In contrast, for three samples, the vertical slide led
to a significant change in D$_e$, but neither analysis version agreed with the expected doses (Fig. 2c). Nevertheless, we
recommend always using the vertical slide correction, given that sensitivity changes are known to occur and can vary in
extent for different samples (e.g., compare Fig. 3e and f).

We also recommend removing of up to the initial ~35 Gy to avoid the initial rise in IR-RF signal seen in most
measurements (here, we removed the initial ~2 Gy). Using the SAR protocol on the two modern samples, such a rejection
lead to slight dose underestimations (resulting in chronologically impossible negative ages), whereas no rejection yielded D$_e$
values of 0.0 ± 0.6 Gy for both samples, matching the expectation (at 2σ). Similar results were found for the MAR analysis.
We expect that the sharp signal peak caused by the initial rise phenomenon biases the fitting algorithm towards 0 Gy, as the
peaks of the natural and regenerative dose curves align. Therefore, we suggest that the better agreement without any
rejection is merely coincidental. Further work would be needed on low-dose samples to explain the observed
underestimation and assess the best number of rejected initial channels. However, at this stage, we caution against the
reliability of IR-RF D$_e$ values below ~100 Gy, due to the possible interference from other RF emissions (see Fig. 4).

Returning to the non-modern samples, the only common feature among the three samples with SAR D$_e$ values that
did not match expected doses was a large increase in D$_e$ (49–116%, with a vertical slide) when using progressively higher-





dose segments of the natural dose curve for sliding (Fig. 3h). The $D_e$ increase of the four other non-modern samples was much lower (12–28%), indicating lower sensitivity changes. We, therefore, tentatively propose using this characteristic as a

screening tool to select samples that can be reliably dated by a SAR IR-RF protocol. Based on the current samples ranging ~100−300 Gy, a threshold of 35% in $D_e$ increase of the final dose segment (~1500−1800 Gy) relative to the initial dose segment (~2−300 Gy) would separate the two groups (Fig. 9). Use of a MAR protocol succeeded in obtaining $D_e$ values compatible with the expected doses for two of the three samples with large progressive sensitivity changes (see Fig. 5a).

We also compared two IR-RF$_{70}$ detection windows (~825–860 nm and ~875–885 nm). In theory, a detection

window further into the IR would be advisable by being less prone to contamination from a red emission. However, for the present samples, no significant accuracy improvement was observed (see Fig. 2c), suggesting that the sensitivity changes occurring during the natural dose curve measurement are not caused by an overlay of an unstable emission. However, the existence of other emissions (or non-radiative recombinations) could be responsible for additional sensitivity changes by altering the proportion of electrons available for the IR-RF process, i.e., through competition effects. Such effects would be

present at any detection window of the IR-RF signal. This finding directly supports the comparability of the new IR-RF results with those that would have been obtained with a Risø system (as used for the room temperature measurements), whose standard IR-RF filter has an effective detection window spanning 850–875 nm. A previous IR-RF$_{70}$ laboratory comparison that included both systems also supports their comparability, having yielded $D_e$ values matching at 1σ (Murari et al., 2021b).

Our comparison of IR-RF$_{70}$ and IRPL results indicated some interesting differences and similarities between the methods. Whatever causes the overestimation of ~25–70 Gy in the IRPL signals for the two modern samples does not seem to affect the IR-RF$_{70}$ signal, despite our understanding that the same trap populations are being accessed. Inversely, the large overestimation of three samples with IR-RF$_{70}$ in the expected dose range 100–160 Gy was not observed for any of the IRPL signals, which yielded $D_e$ values not only closer to the expected doses, but depending on the IRPL signal (see Fig. S10) even

matching the expectations (at 1σ).

Lastly, the DRC shape and signal saturation of the SAR IR-RF$_{70}$, IRPL and pIRIR signals of sample Gi326 were very similar (see Fig. 8), with field-saturation occurring below 85% or below 90% of the full saturation for the SAR and MAR protocols, respectively. This result aligns with the field-saturation of 84% reported by Buylaert et al. (2012b) for the IR-RF$_{RT}$ signal of a different sample. Failure to reach saturation even with a pIRIR$_{290}$ signal is in agreement with the findings

of Yi et al. (2016) when using a high test dose (~500 Gy in their study and also here for the IRPL and pIRIR of sample Gi326). Although we report very similar IR-RF$_{70}$ DRC shapes for all but two samples (Fig. 7b, inset), we note that the aliquots used to obtain these curves are composed of hundreds of grains and previous single-grain investigations have pointed to a larger DRC shape variation, including for a sample also used in the present work (Sontag-González et al., 2024b). The possibility of individual grains with an earlier or later-saturating IR-RF$_{70}$ DRC could explain the variability

observed here for the multi-grain aliquots of some samples.



In comparison with the SAR DRC, the MAR IR-RF$_{70}$ DRC saturated slightly earlier: 85% of the signal intensity (relative to the signal at a dose of ~4000 Gy) was reached after a dose of ~1050 Gy, in contrast to ~1300 Gy in the SAR DRC. Due to decreased sensitivity changes, MAR DRCs are expected to match a natural DRC more closely (e.g. for quartz, Peng et al., 2022). An earlier onset of saturation of the natural IR-RF$_{70}$ signal is also supported by a natural IR-RF$_{70}$ DRC

built by Buchanan et al. (2022) using coarse-grain K-feldspar samples of known age from the Chinese Loess Plateau. Their natural DRC is not incompatible with our MAR DRC (Fig. S11a), though the high scatter in their data does not allow for a definitive comparison (c.f. Fig. S11b).

## 8 Conclusions

The IR-RF age underestimation for samples in the 200–300 Gy range reported by Buylaert et al. (2012b) was likely due to

the measurement temperature (room temperature) and can be overcome by raising it to 70°C (i.e., IR-RF$_{70}$ SAR protocol). Based on our study, we recommend the following: (i) applying the vertical slide correction to account for sensitivity changes, (ii) using a natural dose curve segment length of ~600 Gy for sliding, and (iii) rejecting no more than the initial 35 Gy to mitigate the effects of the initial signal rise. Using these parameters, we observed an agreement with expected ages for four out of seven non-modern samples. However, two modern samples yielded slightly underestimated ages (ca. −2 ka).

Furthermore, we propose implementing a sensitivity change test to identify samples unsuitable for IR-RF dating through the comparison of D$_e$ values obtained by using progressively higher-dose segments of the natural dose curve for sliding. Although this procedure increases the measurement time, it might be a worthwhile addition until we better understand what causes the sensitivity changes during the first irradiation of some samples.

Our preliminary investigations using a MAR method suggest it holds promise, especially for samples with large

sensitivity changes. However, since only six out of the nine samples of known age produced the expected D$_e$ values, we cannot yet recommend its routine application. In our approach, application of the MAR method only required the measurement of one additional aliquot per sample after the traditional SAR measurements.

For samples ranging from ~100−300 Gy, the mean D$_e$ values derived from the pIR$_{50}$IRPL$_{955}$ and the APh-IRPL$_{880}$ signals are in good agreement (at 2σ) with the expected doses, including two samples strongly overestimated with IR-RF$_{70}$.

However, IR-RF$_{70}$ demonstrated better agreement than any of the IRPL signals for the two modern samples. Finally, our study points to a similar upper limit of 1000–1300 Gy for both IR-RF and IRPL, despite the regenerative DRCs of both signals not reaching complete saturation. This finding supports the notion that both techniques target the same traps and indicates that there is some instability of the stored charges.

**Acknowledgements**

The authors thank Jan-Pieter Buylaert for access to data and samples used in a previous study (Buylaert et al., 2012b) and Bo Li for fruitful discussions. This study was supported by the German Research Foundation (MSG and MF: DFG FU417/36-1;



MKM and MF: DFG FU417/19-1). Mette Adrian and Vicki Hansen are thanked for sample preparation. Myungho Kook is thanked for technical support during the IRPL measurements.


**Author contributions**

MSG, MKM, M. Frouin and M. Fuchs designed the experiments and organized the inter-laboratory comparison. MSG and M. Frouin carried out IR-RF measurements. MSG and MJ carried out IRPL measurements. MSG analysed the results and prepared the manuscript with contributions from all authors. M. Fuchs obtained funding.


**Data availability**

The data analysed in this work is available at https://doi.org/10.5281/zenodo.14507180 (Sontag-González et al., 2024a).

**Competing interests**

The authors declare that they have no conflict of interest.

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





**Figures**

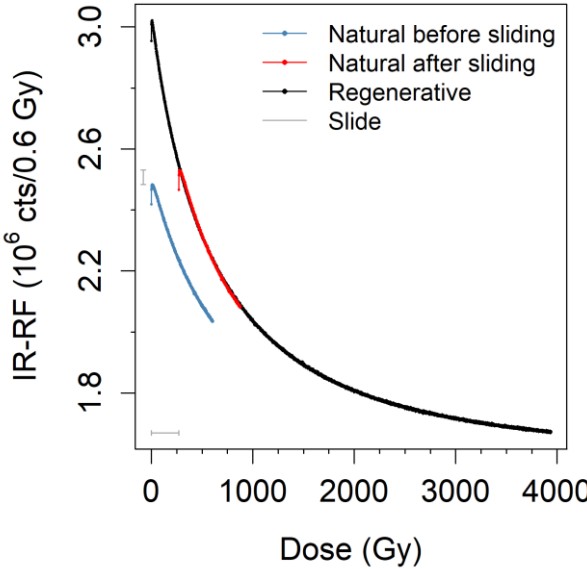

**Figure 1:** *IR-RF$_{70}$ measurements of one aliquot of sample 092202 before and after vertical and horizontal sliding of the natural dose curve. The D$_e$ determined for this aliquot was 270.9 ± 0.5 Gy (without considerations of the uncertainties of the*

*source calibration or the sliding algorithm).*




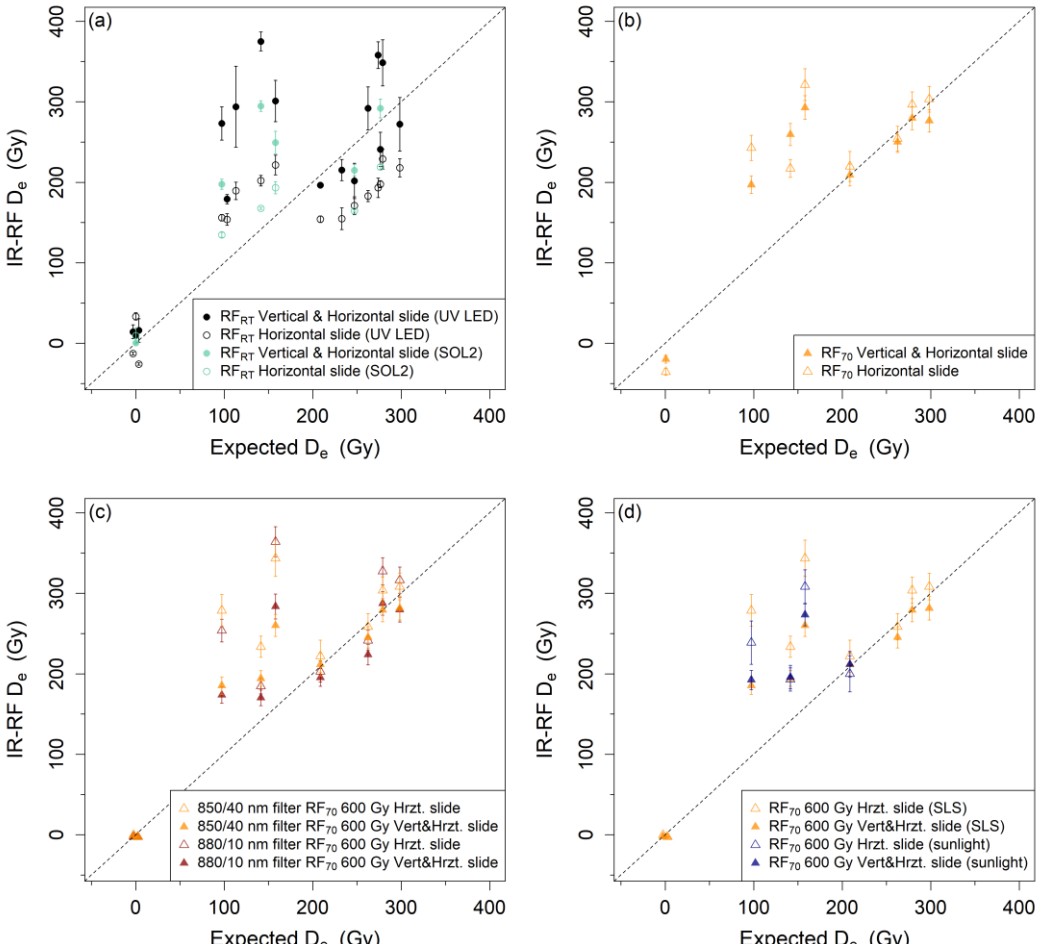

***Figure 2:*** *Comparison of published and new IR-RF results with the expected $D_e$ values. (a) IR-RF was measured at room temperature (RT). The measurements from Buylaert et al. (2012b) (open circles) were analysed without vertical slide. Those measurements were re-analysed here with vertical slide correction of sensitivity changes (filled circles). Aliquots were bleached with UV LEDs or an external solar simulator (SOL2) between the natural and regenerative dose measurements. (b) IR-RF of a subset of the same samples as in the previous panel measured at 70°C. $D_e$ values were obtained with (filled triangles) and without (open triangles) vertical slide. The same natural dose segments as in Buylaert et al. (2012b) were used for sliding. (c) All IR-RF$_{70}$ with different detection filters using a segment of the natural dose curve spanning 600 Gy but rejecting the initial ~3 Gy. (d) Mean $D_e$ values obtained with different bleaching regimes between the natural and regenerative dose steps: the internal solar simulator made up of an LED array (SLS; 9 samples) or direct sunlight (4 samples). $D_e$ values were obtained either with (closed triangles) or without (open triangles) vertical slide. All measurements were made using an 850 nm filter (FWHM: 40 nm). Note that a 5 Gy interval was placed between the expected doses of the modern samples to aid visualization.*







***Figure 3:*** *Comparison of IR-RF$_{70}$ mean $D_e$ values using fixed segments of the natural dose curve and (a, c, e, g) only horizontal sliding or (b, d, f, h) vertical and horizontal sliding. For clarity, the used segments of a representative natural dose curve (sample 092202) are shown in the legend box to the right of the corresponding plots. Note that a 5 Gy interval was placed between the expected doses of the two modern samples to aid visualization. The $D_e$ for the field-saturated sample Gi326 is shown on the right-hand y-axis in each plot; its expected $D_e$ is 'saturated' (sat.). The dashed line indicates the 1:1*

*line. (e, g) Arrows indicate a minimum estimate caused by the limit of the regenerative dose curve.*





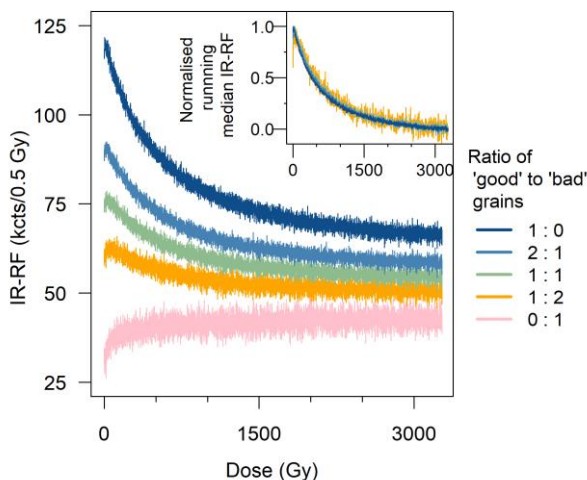

***Figure 4:*** *Simulation of regenerative dose IR-RF$_{70}$ curves of aliquots of sample H22550 composed of a combination of grains with the expected IR-RF signal shape ('good' grains, Type #1) and grains with an initially increasing signal shape*

*('bad' grains, Type #2). The 'good' grain curve (dark blue) is the average of three individual grains with the expected signal shape, whereas the 'bad' grain curve was obtained from one grain. The grains are referred to as ID #3, #4, #5 and #6 in Sontag-González et al. (2024b). The inset shows the same curves (except for that composed entirely of Type #2 grains) normalised to their maximum signal values, removing as background the median value of the last 100 channels (~60 Gy). Note the decrease in 'initial rise' with increasing proportion of 'good' grains relative to the total dynamic signal range.*






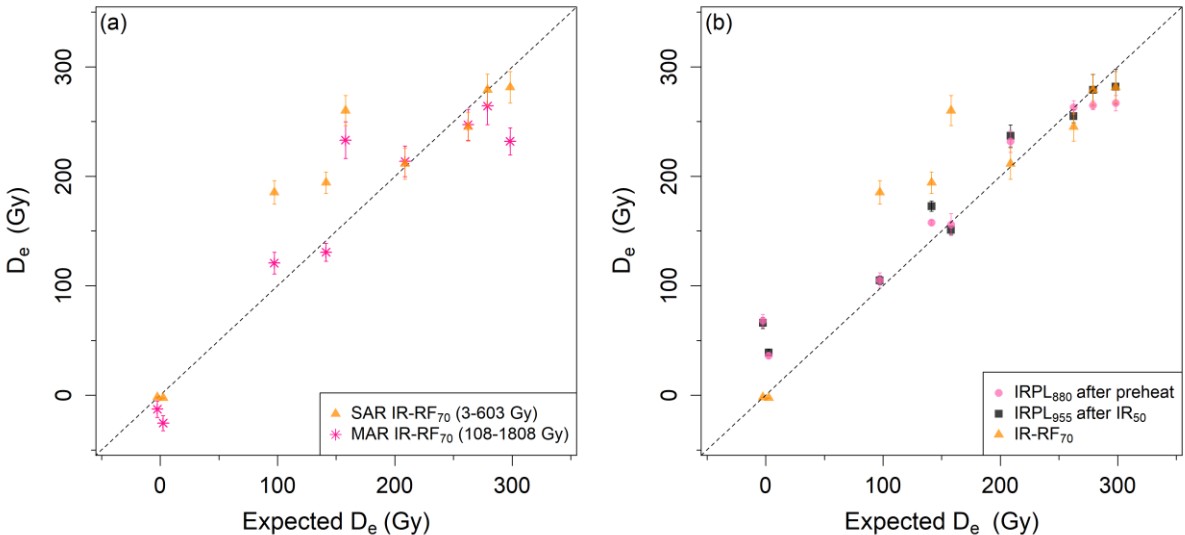

***Figure 5:*** *Comparison of the new IR-RF$_{70}$ and IRPL results with the expected D$_e$ values. (a) Single-aliquot regenerative dose (SAR) IR-RF$_{70}$ results and those obtained using a multiple aliquot regenerative dose (MAR) protocol. Results are shown for*

*the most reliable natural dose curve segment of each protocol, as given in the legend. (b) IRPL and SAR IR-RF$_{70}$ measured on of the same set of samples. The IRPL D$_e$ are a combination of new measurements and those from Kumar et al. (2021). The subscript after 'IRPL' in the legend indicates the wavelength of the targeted emission. IR-RF$_{70}$ D$_e$ values were obtained with vertical and horizontal slide using a segment of the natural dose curve spanning 600 Gy but rejecting the initial ~3 Gy. Note that a 5 Gy interval was placed between the expected doses of the modern samples to aid visualization.*




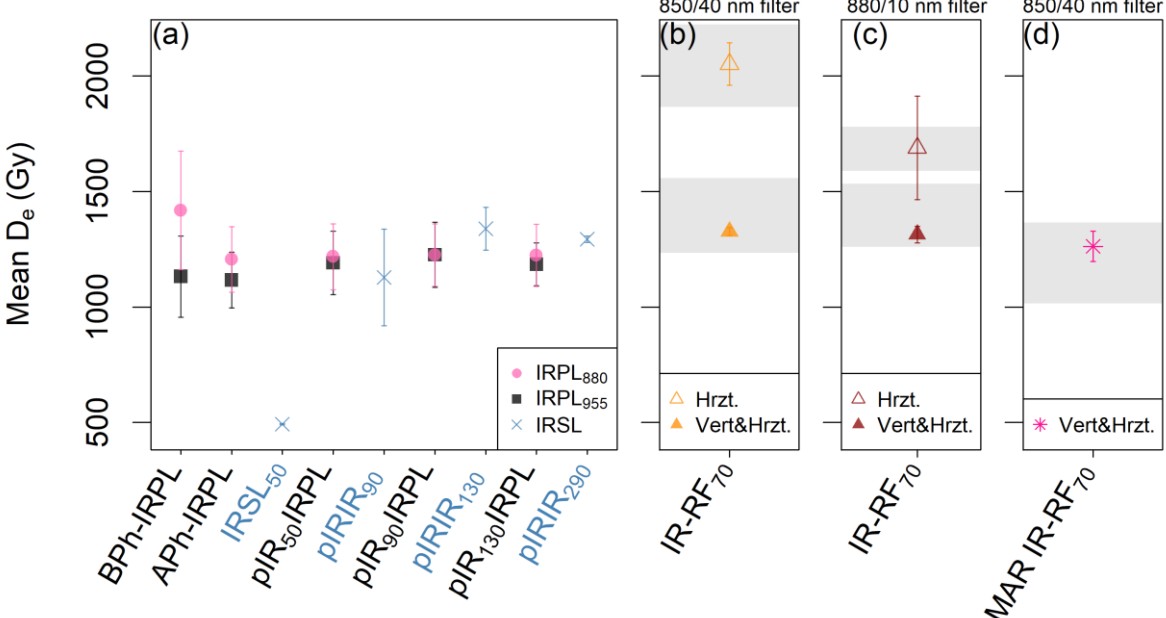

***Figure 6:*** *Comparison at field-saturation for sample Gi326. (a) Mean $D_e$ values obtained from the various steps in the IRPL sequence, including the IRSL steps (highlighted in blue in the x-axis labels). 'BPh' and 'APh' refer to before and after the preheat step, respectively. The IR-RF mean $D_e$ values were obtained using (b, d) an 850/40 nm and (c) an 880/10 nm bandpass filter. (b, c) The initial ~600 Gy segment of the natural dose curve (rejecting the first 3 Gy) was used with horizontal or with vertical and horizontal sliding (Hrzt. and Vert&Hrzt., respectively). (d) Results obtained using a multiple aliquot regenerative dose (MAR) protocol and sliding the natural dose curve segment spanning 108−1808 Gy. The range of $D_e$ values that could be obtained using alternative natural dose curve segments (i.e., those in (b) Fig. 3e–f, (c) Fig. S6e–f and (d) Fig. S8b–d) are shown as grey bands without the associated uncertainties.*



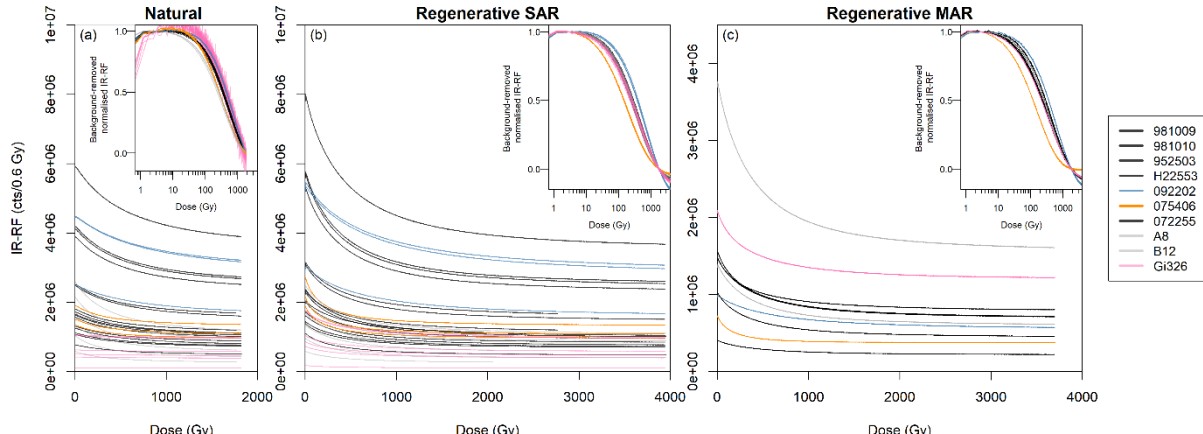

**Figure 7:** *IR-RF DRC shape comparison of (a) the natural, (b) the regenerative dose signal using a SAR and (c) the regenerative dose signal using a MAR protocol; all measured with an 850 nm bandpass filter (FWHM: 40 nm). In the insets, a background was subtracted from the IR-RF curves, corresponding to the signal value at a dose of 1800 Gy before normalisation to the signal value at 3 Gy. Curves from the two modern samples and the field-saturated sample are shown in pink and grey, respectively. The DRCs of two additional samples are coloured to highlight their deviation from the curve shape of all other samples.*



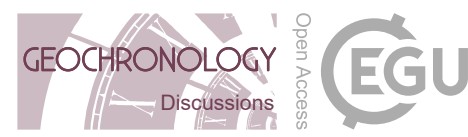


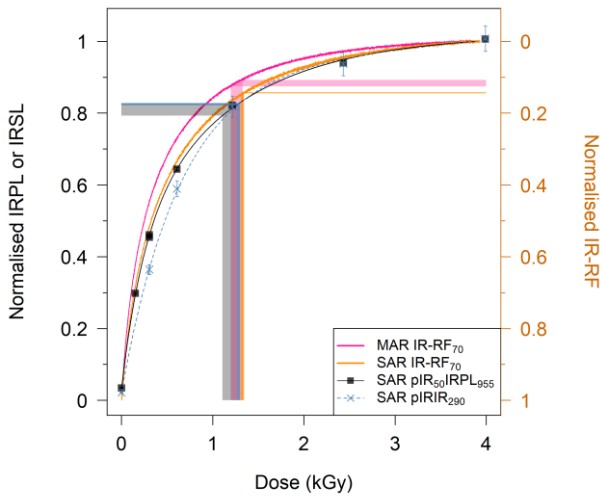

***Figure 8:*** *DRC shape comparison of representative aliquots of Gi326. The pIR$_{50}$IRPL$_{955}$ and the pIR$_{90}$IR$_{290}$ curves are double exponential functions fitted to the sensitivity corrected signals (L$_x$/T$_x$) of 7–8 regenerative doses each, including a repeated dose point (~300 Gy). All curves were normalised to their maximum values. Before normalisation, a background*

*was subtracted from the IR-RF curves, corresponding to the median value of the last 100 channels (~60 Gy). The IR-RF curves decrease with dose, so they were plotted on an inverted axis (right-hand side) to allow for a direct comparison. The shaded regions indicate the 1σ range around the four mean D$_e$ values and their projection onto the corresponding DRC.*




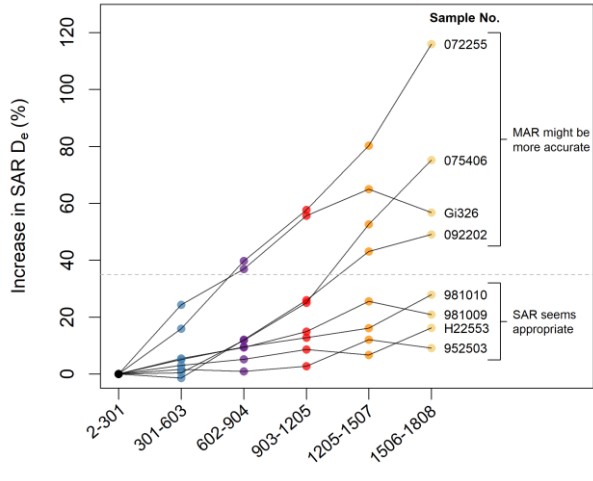

***Figure 9:*** *Test of progressive sensitivity changes during the IR-RF$_{70}$ natural dose curve measurement. $D_e$ values of non-modern samples were obtained using fixed segments of the natural dose curve according to the x-axis labels with vertical and horizontal sliding (same data and colours as in **Fig. 3h**). The increase in $D_e$ is relative to that obtained from the earliest*

*segment (2−301 Gy). The dashed line (35% increase) separates samples that produce $D_e$ values within 2σ of the expected dose and those that do not.*





**Tables**

***Table 1:*** *Details on sampling locations and previous dating studies. 'se' refers to the standard error at 1σ. IR-RF$_{RT}$ refers to*
*the room temperature measurement protocol. Previous IR-RF$_{RT}$, IR-RF$_{70}$ and IRPL ages are given in Buylaert et al. (2012b), Murari et al. (2021b) and Kumar et al. (2021), respectively.*

| Code | Site & Location | Sediment type | Grain size (µm) | Expected Age (ka) | Type of age control | Previous IR-RF or IRPL studies |
|---|---|---|---|---|---|---|
| 981007 | Gammelmark (Denmark) | coastal marine sand | 150–250 | 128±2 | Biostratigraphy and quartz OSL (Murray and Funder, 2003; Buylaert et al., 2012a) | IR-RF$_{RT}$ |
| 981009 | | | 150–250 | 128±2 | | IR-RF$_{RT}$, IRPL |
| 981010 | | | 150–250 | 128±2 | | IR-RF$_{RT}$, IRPL |
| 981013 | | | 90–150 | 128±2 | | IR-RF$_{RT}$, IRPL |
| 952503 | Sula (Russia) | coastal marine sand | 150–212 | 128±2 | Biostratigraphy and quartz OSL (Murray et al., 2007; Buylaert et al., 2012a) | IR-RF$_{RT}$, IRPL |
| H22548 | | | 180–250 | 128±2 | | IR-RF$_{RT}$ |
| H22550 | | | 180–250 | 128±2 | | IR-RF$_{RT}$ |
| H22553 | | | 180–250 | 128±2 | | IR-RF$_{RT}$, IRPL |
| 092202 | Indre-et-Loire (France) | colluvium | 180–250 | 45.5±2.5 | [14]C (Aubry et al., 2012) | IR-RF$_{RT}$, IRPL |
| 075403 | Sinai peninsula (Egypt) | reworked dune sand | 180–250 | 34±2 | | IR-RF$_{RT}$ |
| 075406 | | | 180–250 | 38±2 | | IR-RF$_{RT}$ |
| 075407 | | | 180–250 | 37±2 | | IR-RF$_{RT}$, IRPL |
| 072255 | Carregueira (Portugal) | aeolian sand | 180–250 | 20.2±1.2 | Quartz OSL (Buylaert et al., 2012a) | IR-RF$_{RT}$, IRPL |
| 055642 | North Jutland (Denmark) | beach sand | 90–180 | -0.0007±0.0017 | | IR-RF$_{RT}$ |
| 102011 | Maria Pia Beach (Sardinia) | beach sand | 180–250 | -0.001±0.009 | | IR-RF$_{RT}$ |
| A8 | Dune, Rømø (Denmark) | aeolian dune sand | 180–250 | 0.38±0.029 | Quartz OSL (Madsen et al., 2007) | IR-RF$_{RT}$ |
| B12 | Wadden Sea (Denmark) | coastal marine deposits | 180–250 | 0.01±0.002 | | IRPL |
| Gi326 | Bayreuth, Germany | Triassic sandstone | 90–200 | ~250 Ma | Lithostratigraphy (Murari et al., 2021b) | IR-RF$_{70}$ |





***Table 2:*** *IR-RF measurement protocols. 'Natural dose' curve (step 2) refers to the RF measurement where an additive dose is administered on top of the natural dose. The length of the 'regenerative dose' measurements (step 6) was sometimes shortened for younger samples.*


| Step | IR-RF$_{RT}$ SAR protocol (Buylaert et al., 2012b) | IR-RF$_{70}$ SAR protocol | IR-RF$_{70}$ MAR protocol | | Purpose |
|---|---|---|---|---|---|
| | | | Natural (same as SAR) | Regenerative | |
| 1 | - | Preheat at 70°C for 900 s | Preheat at 70°C for 900 s | - | Stabilise temperature |
| 2 | RF at room temperature for between ~9 Gy and ~480 Gy | RF at 70°C for 30 000 s (~1800 Gy)* | RF at 70°C for 30 000 s (~1800 Gy)* | - | Obtain 'natural dose' curve |
| 3 | UV bleach for 1500 s or Hönle SOL2 bleaching for 14 400 s | 'Solar simulator' bleaching for 24 817 s | 'Solar simulator' bleaching for 24 817 s | 'Solar simulator' bleaching for 24 817 s | Fully remove signal |
| 4 | Pause for at least 1 h | Pause for 2 h | Pause for 2 h | Pause for 2 h | Reduce phosphorescence |
| 5 | - | Preheat at 70°C for 900 s | Preheat at 70°C for 900 s | Preheat at 70°C for 900 s | Stabilise temperature |
| 6 | RF at room temperature for between ~350 Gy and ~2940 Gy | RF at 70°C for between 35 000 and 65 000 s (~2100–3900 Gy) | RF at 70°C (at least one channel) | RF at 70°C for 65 000 s (~3900 Gy) | Obtain 'regenerative dose' curve |

*\* Only a segment of the natural dose curve was used for final D$_e$ estimation, see section 3.3.*





*Table 3:* *Results from IR-RF measurements at room temperature (RT) either with or without sensitivity change correction. Data has been re-analysed from Buylaert et al. (2012b). Aliquots were bleached with UV LEDs or an external solar*
*simulator (SOL2) between the natural and regenerative dose steps. 'n' refers to the number of aliquots used to determine the arithmetic mean $D_e$.*

| Sample code | Expected $D_e$ (Gy) | IR-RF$_{RT}$ (UV LED bleach) | | | IR-RF$_{RT}$ (SOL2 bleach) | | |
|---|---|---|---|---|---|---|---|
| | | n | Horizontal slide $D_e$ (Gy) | Vertical and horizontal slide $D_e$ (Gy) | n | Horizontal slide $D_e$ (Gy) | Vertical and horizontal slide $D_e$ (Gy) |
| 981007 | 276.5±11.1 | 6 | 197.7±4.5 | 240.9±21.3 | 4 | 218.9±3.2 | 291.9±11.7 |
| 981009 | 279.0±11.1 | 11 | 229±12.5 | 348.4±28.6 | - | - | - |
| 981010 | 298.2±12.4 | 4 | 218±11.3 | 272.1±33.3 | - | - | - |
| 981013 | 273.9±12.3 | 5 | 193.2±11.9 | 358±16.6 | - | - | - |
| 952503 | 262.4±12.2 | 11 | 182.9±6.9 | 291.9±26.7 | - | - | - |
| H22548 | 247.0±9.8 | 5 | 171±11.1 | 201.7±21.9 | 4 | 164.5±1.3 | 214.9±7 |
| H22550 | 233.0±10.9 | 10 | 154.9±13.5 | 215.2±13.1 | - | - | - |
| H22553 | 208.6±10.7 | 3 | 154.3±4.1 | 196.4±3.3 | - | - | - |
| 092202 | 157.9±10.5 | 13 | 221.4±12.3 | 301±25.6 | 4 | 193.3±7.4 | 249.4±14 |
| 075403 | 113.2±8.2 | 6 | 189.4±10.8 | 293.9±50.3 | - | - | - |
| 075406 | 141.4±9.6 | 9 | 202.1±6.6 | 374.9±11.7 | 4 | 167.8±1.4 | 294.7±6.5 |
| 075407 | 103.2±6.9 | 3 | 153.9±7.3 | 179.1±5.7 | - | - | - |
| 072255 | 97.2±6.8 | 15 | 156.2±4.1 | 273.2±20.6 | 4 | 134.8±3 | 197.7±6.3 |
| 055642 | -0.002±0.004 | 6 | -12.6±1.5 | 14.3±8.6 | - | - | - |
| 102011 | -0.01±0.01 | 8 | 33.4±4.5 | 9.8±2.2 | 4 | 9.7±9.7 | 0.5±1.4 |
| A8 | 0.64±0.06 | 6 | -25.7±1.4 | 16.1±14.5 | - | - | - |



***Table 4:*** *Results from IR-RF measurements at 70°C using different bandpass filters and IRPL measurements after the preheat step (APh) and after the IRSL$_{50}$ step (pIR$_{50}$IRPL). IR-RF D$_e$ values were obtained using either a single or*

*multiple aliquot regenerative dose (SAR or MAR, respectively) procedure. SAR IR-RF D$_e$ values were obtained using the initial ~600 Gy of the natural dose curve, rejecting the initial ~3 Gy and applying a sensitivity change correction (horizontal and vertical slide), whereas MAR D$_e$ values were obtained from segments of the natural dose curve spanning 100−1808 Gy. IRPL measurements marked with an asterisk are taken from Kumar et al. (2021). 'n' refers to the number of aliquots used to determine the arithmetic mean D$_e$.*

| Sample code | Expected D$_e$ (Gy) | IR-RF$_{70}$ | | | | | IRPL | | | | |
| | | 850/40 nm filter | | | 880/10 nm filter | | | IRPL$_{880}$ D$_e$ (Gy) | | IRPL$_{955}$ D$_e$ (Gy) | |
| | | n | SAR D$_e$ (Gy) | MAR D$_e$ (Gy) | n | SAR D$_e$ (Gy) | n | APh | pIR$_{50}$IRPL | APh | pIR$_{50}$IRPL |
|---|---|---|---|---|---|---|---|---|---|---|---|
| 981009 | 279.0±11.1 | 4 | 279.1±14.4 | 264.3±17.1 | 3 | 287.5±14.8 | 3* | 265±4 | 297±9 | 226±3 | 279±14 |
| 981010 | 298.2±12.4 | 3 | 281.4±14.5 | 232.0±12.4 | 3 | 279.5±15.1 | 3* | 267±7 | 280±5 | 246±9 | 282±16 |
| 952503 | 262.4±12.2 | 3 | 245.3±13.2 | 247.1±14.2 | 3 | 223.8±12.6 | 6* | 263±6 | 288±7 | 226±8 | 255±6 |
| H22553 | 208.6±10.7 | 3 | 211.5±14.2 | 213.6±14.0 | 3 | 195.5±11 | 3* | 232±10 | 248±11 | 205±10 | 237±10 |
| 092202 | 157.9±10.5 | 3 | 260.1±13.7 | 233.0±16.7 | 3 | 283.7±15.4 | 3* | 156±10 | 159±11 | 146±3 | 151±4 |
| 075406 | 141.4±9.6 | 3 | 194.2±9.8 | 130.6±8.2 | 3 | 170.2±9.8 | 4 | 158±2 | 176±4 | 139±3 | 173±5 |
| 072255 | 97.2±6.8 | 3 | 185.3±10.7 | 120.7±9.9 | 3 | 173.7±10.3 | 3* | 105±7 | 103±6 | 94±2 | 105±4 |
| A8 | 0.64±0.06 | 3 | −2.4±0.6 | -25.6±6.9 | 3 | −2.3±0.6 | 7 | 36±3 | 41±3 | 24±2 | 39±3 |
| B12 | 0.01±0.01 | 3 | −2.0±0.6 | -12.7±7.6 | 3 | −2.3±0.6 | 3* | 68±6 | 70±4 | 39±1 | 66±5 |
| Gi326 | ~500 000 | 6 | 1327±18 | 1263±65 | 5 | 1314±36 | 3 | 1206±141 | 1217±143 | 1116±120 | 1191±138 |
