# Peer review of "Further investigations into the accuracy of infrared-radiofluorescence (IR-RF) and its inter-comparison with infrared photoluminescence (IRPL) dating"

_Geochronology, 2024_

## Author Response (AR1)

**RC1**

➔ We thank the reviewer for their positive assessment of our work. We have responded to the raised issues below, where the original comments are in grey font and the responses in black prefaced by an arrow.

**Summary**

Over the past two decades, two groundbreaking luminescence new dating techniques have emerged: infrared radiofluorescence (IR-RF) on feldspar specimens and infrared photoluminescence (IR-PL). Although IR-RF is the older method, it has only recently gained more adoption. Since the introduction of IR-PL, there has been ongoing debate regarding its superiority over IR-RF. The primary advantage of both techniques lies in their ability to measure the charge density of a principal trap. This measurement provides a more direct estimate of the accumulated dose.

The manuscript presents in essence a long-overdue systematic comparison of IR-RF and IR-PL that should have been done long ago on a set of samples. The present contribution goes even beyond as it also re-analysed older results from a study that has questioned the reliability of IR-RF.

The approach is systematic, and the manuscript is carefully prepared. It takes a somewhat neutral standpoint, although most of the tests are clearly related to IR-RF. Nevertheless, what is refreshing is that it does not claim superiority of one method over the other but tries to point out differences and challenges where applicable.

The manuscript clearly aligns with the scope of Geochronology and should be published.

I have only a few general remarks and more minor technical comments, but I am confident that they can be addressed by the study authors easily.

**General remarks**

- The manuscript is generally well-structured and well-written, with most sections being easy to read. However, I had the impression that the authors added more experiments along the way and somewhat forgot the original purpose of the manuscript. While this is common, I suggest that the authors revise the introduction to make it easier for readers by clearly marking hypotheses that can be quantified and tested. For instance, the title suggests a straightforward comparison of IR-RF and IRPL, but the study then presents a diverse range of tests (including pIRIR measurements). In other words, the study lacks some rigour and could benefit from a little streamlining.

➔ We have included the IRSL/pIRIR aspect in the introduction, though we highlight that these results are taken directly from the IRPL protocol of Kumar et al. (2021) and not measured separately. Though our focus was on the IRPL results, we chose to also report the IRSL ones, since we had the data.

➔ We have also rephrased the last paragraph of the introduction to make the hypotheses more explicit.

- The MAR test is intriguing, but it comes unexpectedly and lacks further substantiation. This presents a missed opportunity. The authors should either extend this section or remove it, as it will not significantly alter the outcome but blur the story. By doing so, they would also have the opportunity to design more robust experiments and incorporate modelling.

➔ We agree that the IR-RF MAR results are preliminary and that more work still needs to be done. However, we believe that testing a MAR approach follows logically from the reported issue of uncorrected sensitivity changes in the SAR. We have chosen to retain the section to serve as a motivation for future research. We state in the conclusion that the MAR results are preliminary and have now expanded on the reasons behind the inclusion of a MAR approach in the introduction, so it does not appear as unexpectedly.

- The same goes for the discussion about the detection window. This is given quite some attention, but at the end, it seems to have no effect. If so, this can be shortened to two or three lines and with the rest of the data in the supplement.

➔ Agreed, we have removed the detailed descriptions from the main text and kept the results in Fig. S8 and Table 4.

- The authors provide access to raw data, which is highly appreciated; however, they should also add full information on the source rate calibration (see below). Otherwise, the results are of limited use to others.

➔ We have added the dose rate calibration details used for the new measurements. For the data from Buylaert et al. (2012), the dose rate used for each aliquot is provided in the Zenodo dataset under 'RLanalyse_De.csv'.

- The definition of uncertainties appears somewhat ambiguous, often I am not sure whether they are truly comparable. For instances, sometimes it seems to be the standard deviation, sometimes the standard error of the mean, for the independent age control something else(?).

➔ The stated uncertainties for our mean $D_e$ estimates are the standard error of the mean. This metric was chosen to remain comparable with the work of Buylaert et al. (2012) and the independent ages reported therein, whose uncertainties according to their Table S1 represent one standard error. In the few cases in which the standard error of the mean was below the channel length (10 s, approximately 0.6 Gy), we considered the channel length as the mean $D_e$ uncertainty because that is the limit of our resolution. We have added this information to section 2.3. To avoid confusion, we have removed the $D_e$ estimate of one aliquot from the caption of Fig. 1, whose uncertainty (the standard deviation of a Monte Carlo simulation of the $D_e$ distribution

bootstrapping the residuals of the curves after sliding) was not used for later calculations.

**Minor comments**

- L37: Because you explain basics, you should also addd a suitable reference for IRSL

➔ Agreed, we have added a reference.

- L38: The reference to Krbetschek et al. (2000) seems incorrect. The authors wrote "Fading tests (storage over periods of several months at room temperature) have shown signal stability." Krbetschek et al. (2000, p. 497). They further stated: "Further investigation is necessary to ascertain what this tells us about the mean life of the trap population" (2000, p. 497). They did not write anything about "lower anomalous fading rates". They wrote about signal stability and the mean life(time) of the trap population. This implies that they meant the thermal not the athermal lifetime.

➔ We interpreted the result of "signal stability" (Krbetschek et al., 2000; p. 497) obtained from the fading test as a fading rate consistent with zero, which is lower than the usual non-zero rates obtained for IRSL. In any case, we have rephrased this sentence to "a more athermally stable signal".

- L46: Technically, the sliding approach goes back to Prescott et al. (1993) (or even earlier) under the name "Australian slide". Buylaert et al. (2012) describe horizontal sliding in their Fig. 4; the method/tool is described in Lapp et al. (2012) where they describe a time-shift. Kreutzer et al. (2017) (see also in Murari et al. 2018) first used the approach; Murari et al. (2018) formally introduced it. However, Buylaert et al. (2012) indeed mention horizontal and vertical adjustments, but it is unclear what their conclusion was and why they did not test it. But I agree, credit should be given to them because they mention the idea.

➔ Agreed, we have rephrased and added the reference Kreutzer et al. (2017).

- L86-L95: This paragraph is very muddled. You start re-analysing 16 samples but present 10 new IR-RF and then again "eight samples originally used". Please rephrase to improve readability or make a list for your experiments or a workflow graph.

➔ We have rephrased the paragraph and added a supplementary overview figure showing how our chosen set of samples relates to those measured in previous studies for clarification.

- L86-L95: The introduction should explicitly state a research hypothesis that will be tested in the contribution, rather than presenting a list of experiments that may or may not yield a specific outcome; some of them even unrelated to the study title.

➔ The hypothesis behind each of the tests is that varying that parameter will lead to more accurate IR-RF $D_e$ values. We have rephrased the final paragraph of the introduction to clarify this and name the tested parameters, e.g., the use of the vertical slide, the length of the natural dose curve used for sliding, the number of rejected initial channels, the detection window and the type of bleach between the natural and regenerative dose curve measurements.

- L102-L103: The HF etching of feldspars is challenging, and still it is unlikely to remove any other alpha-irradiated layer uniformly and in the desired manner (Duller, 1992; Porat et al., 2015; Duval et al., 2018). Given that sample preparation cannot be altered retrospectively, I propose discussing this issue and its potential impact on the final results later in the manuscript.

➔ We have commented on this issue in section 2.1, but since all samples expected to be in the dating range received the same treatment, we do not expect etching to affect our conclusions.

- L116-L125: Please provide detailed information on the calibration of the other machines, including the aliquot size, sample carrier, dose rate, and calibration date. This information is essential for cross-checking your results by others without the need for additional inquiries. If necessary, please indicate whether you had to correct calibrations based on previous measurements (as described in Autzen et al., 2022). I attempted to recalculate a few results using the data provided on Zenodo. However, without the dose rate (available for some samples, though), I am unable to effectively compare the results.

➔ We have added the dose rate calibration details for the missing readers used for the new measurements.
➔ We have now corrected all measurements taken from Buylaert el al. (2012) using the 8.25% value suggested by Autzen et al. (2022). We also used this value to correct the control ages that had been estimated using OSL.

- L136: Please state the number of ignored channels, or the dose (you do that later). I tried to recalculate, for instance, A8. But my result is consistently 0, however, I can get any result (also the one you report) by ignoring a certain number of channels. Means, this information matters.

➔ We did not state the number of rejected channels here in the methods section because this was a parameter we varied. We have added a sentence stating that between 0 and 499 channels were ignored, which will be detailed in a later section. In Table 4, where we present results from the optimal tested parameter combination, we detail in the

caption that 2 Gy were removed. We have added the number of ignored channels to the caption of Table 4 as well as to the discussion.

- L139: A more detailed description with of the initial rise can be found in Frouin et al. (2017) (their supplement). From this analysis, it becomes evident that the response appears to be dose-dependent and exhibits a range of responses within a given dose range.

➔ We now refer to the initial rise results from Frouin et al. (2017).

- L163: What is the justification for the double-exponential fit? Wouldn't the GOK model (Guralnik et al., 20215) be a better candidate for feldspar?

➔ We used a double exponential to be able to directly compare our results with those from Kumar et al. (2021). We do not expect that a GOK model would change the results significantly.

- L176: Can you show such a distribution?

➔ We have added example $D_e$ distributions (850/40 nm filter) as a supplementary figure.

- L215: I think that the number of channels matters more than the dose; please also check the supplement by Frouin et al. (2017) where this investigated (although it seems only with horizontal sliding).

➔ We have checked the suggested investigation by Frouin et al. (2017), who undertook an analysis of incrementally increasing the segment length up to 100 channels (horizontal slide). Our results are compatible with those presented there when we apply only the horizontal slide, as shown below for sample H22553 (always rejecting the first 4 channels). Looking only at the initial 100 channels (Fig. inset), it appears that a plateau is reached after 60 channels (~36 Gy). However, if continuing the analysis, the mean $D_e$ rises again and continues to change until ~2500 channels (~1500 Gy), then reaching a plateau, which extends until at least 3000 channels (~1800 Gy). The early plateau is reached here at a similar dose as in the study of Frouin et al. (2017), who report >40 channels (~30 Gy with their 0.7 Gy/channel) are needed to reach a $D_e$ plateau. Overall, it seems that a segment length of <100 channels does not yet tell the whole story. We now refer to the results from Frouin et al. (2017) in a new section "Considerations on the IR-RF DRC".

[Figure]

➔ To answer whether the channel length or dose is the more relevant measure, we have re-measured one aliquot of sample 072255 changing the channel length from 10 s to 1 s. These settings serve as a proxy to running the sequence on a reader with a dose rate 10x lower. As shown below, the mean $D_e$ follows the same pattern when looking at the dose (panel a), but not so when looking only at the channels (panel b). This suggests that the cause of the pattern is a dose-dependent characteristic. We have added the number of channels in some key places in the manuscript for reproducibility, but highlight that users would need to adjust measurement parameters (e.g., curve length) to obtain equivalent results when using devices with different dose rates.

[Figure]

- L430: I concur with this conjecture, albeit with a slight distinction. I think that the relationship lies not solely with the dose but also with the number of channels. Your objective should be to identify a plateau of equivalent dose values rather than distinct segments. To achieve this, you can segment your natural dose and incrementally add channels to the RF natural dose until a plateau is detected. While this approach addresses the issue of channel-related variations, it still presents a challenge: if the regenerated and natural curves indeed differ, the results may be inaccurate when compared to an independent age control.

  Nevertheless, this approach eliminates the possibility of arbitrary channel selection.

➔ See comment above on the issue of channels vs. dose.

➔ We have tested the reviewer's suggestion of running the segment length comparison incrementally adding one channel as opposed to our original approach of selecting six segment lengths. Results from one sample are shown below for (a) horizontal and (b) horizontal and vertical slide. Despite testing with a relatively long natural dose curve of 1800 Gy (3000 channels), we do not yet see a $D_e$ plateau with the horizontal and vertical slide. In fact, there is an initial decrease (possibly related to the sliding algorithm) and then an increase. For this sample, the $D_e$ is beginning to stabilize around 1800 Gy, but for other samples the increase is still significant at the same segment length. We have chosen six segment lengths (including a new one not investigated in the original manuscript) that cover the range of likely $D_e$ values (magnification of the region of interest given in the inset). The chosen lengths are also shown in the figure below (coloured points).

[Figure]

- L440-L443: They yield 0 Gy because the algorithm has no other choice to match the curves given the shapes and the starting points and then sets it to 0. This is not coincidental; this is by design. See your own arguments a few lines below.

➔ We meant that it is a coincidence that when not rejecting any initial channels the expected dose is obtained. The agreement should be considered to be an analytical artefact (due to the initial rise always being at the start of the curves) and not evidence of accuracy. For this reason, we recommend always rejecting initial channels even though that leads to a worse accuracy in the case of the modern samples. We have re-phrased the sentence for clarity.

- L437: This somewhat contradicts your chain of arguments trying to emphasise good arguments and put more weight on one or the other. The 35 Gy is an arbitrary choice and sample dependent; it seems a good fit for your samples, but I suggest refraining from generalising this observation. The best approach seems to reject the very first channel and keep the rest (with a certain number of minimum channels)

➔ We did not recommend rejecting 35 Gy, instead, since for some samples the $D_e$ is not stable when rejecting more than ~35 Gy, no more than that should be rejected. In fact, for most samples, rejecting much less is sufficient, as evidenced at the end of this sentence, where we state we rejected only ~2 Gy. We have rephrased for clarity.

- L461-L464: In Murari et al. (2021) all measurements (Risoe and Freiberg readers) seem to have been used 70ºC as recording temperature; please rephrase or remove.

➔ We were referring to the room temperature RF measurements of Buylaert et al. (2012), but we have removed the indication of temperature in the sentence for simplicity.

- L465-L470: Agreed, but you should also point to the different protocols with no less than 17 to 18 steps. I am wondering how sensitive the equivalent dose is to certain parameters. If you cannot test this, you should at least discuss it.

➔ Agreed, added to discussion.

- L483-L484: I do not believe that the comparison to the quartz model is valid. While the observation may share some similarities, the underlying mechanism is unknown and likely distinct in quartz. Unless you can provide a model and demonstrate that the mechanism is indeed similar, I recommend removing this speculative comparison. The subsequent comparison is more appropriate and logical, although it is purely descriptive.

➔ Agreed and removed, though we added a sentence on previous K-feldspar MAR studies.

**Figures**

- Figure : The figure I am missing is a distribution plot for equivalent doses. Perhaps this can be added for suitable samples.

➔ Added as supplementary figure.

- Figure 1: Please add information on the aliquot number and which measurement window was used. Also, here contrary to what was written in the M&M section no initial channel was discarded.

➔ We have added the requested information and a note on the channels used for sliding to the caption. In this case, the initial 4 channels were discarded for sliding. Since the

aim of this figure was to introduce the sliding procedure to readers unfamiliar with it, for simplification, we did not use different colours for the discarded channels. The concept of rejecting certain channels is later introduced graphically in Fig. S6.

- Figure 2: Please colour-code the samples and use shapes to denote the methods. This will prevent readers from having to guess which sample is shown. If you run out of distinguishable colours, please use labels. For the 600 Gy exposure, the quantity of channels is the primary parameter of concern, rather than the dose (the information remains beneficial regardless).

➔ We have changed this figure using colours to distinguish the samples but have retained a method-based colour scheme in Fig. 6, since the number of samples is lower there, allowing them to be distinguished by their dose (or by comparison with the new figure version).

➔ Given the relatively long segments we are working with, we don't expect the observed differences in $D_e$ to be caused by the sliding algorithm (in which case the number of channels might be the primary parameter), but to represent true sample characteristics related to the dose-response, i.e., irrespective of channel length or reader dose rate. See also our reply to the comment on line 215. However, for completion, we have added the number of channels to the caption.

- Figure 3: What is the central new information conveyed by these figures? The sliding method, particularly requires offsetting for short segments and less curvature. Please condense to a single key figure with a succinct message.

➔ This figure is intended to be descriptive and helpful for readers who prefer a visual representation rather than, e.g., tabular data presentation. The main messages described in section 3.3 (now section 4.4) are (i) depending on the segment used for sliding, there can be variation in the resulting mean $D_e$ and (ii) this behaviour is sample-dependent. The wider implications are then summarized in Fig. 10: for samples with small changes in $D_e$, the conventional SAR approach can be expected to yield accurate $D_e$ values. We have added a second panel to Fig. 10 summarizing the change in $D_e$ for increasing segment lengths.

- Figure 5: To compare IR-RF and IRPL, it is necessary to include a third figure that compares both techniques with the method you believe performs most effectively. Additionally, you should compare the distribution of the relative residuals from the 1:1 regression line to assess whether there is a significant difference between the two methods or if they are merely random.

➔ We have added a figure using the APh-IRPL$_{880}$ results in the x-axis.

➔ We have tested the relative residual comparison suggested by the reviewer (for the non-modern samples that have expected ages). The distributions are relatively similar for the three methods, however, we believe that the sample size is too small to reach a definitive conclusion especially considering possible dose-dependent differences. For this reason, we would not include it in the manuscript but only in the supplement.

- Figure 6: Do the grey bands make sense? The optimal range would be with the highest number of channels. However, if you wish to retain the current settings, you should also experiment with different integration values for the other methods. But I guess then it becomes very confusing.

➔ We agree with the reviewer that the equivalent comparison would also include the integration values for IRPL/IRSL, but also think that this inclusion would make the already complex figure somewhat convoluted. We included the grey bands to highlight the subjectivity of our chosen number of channels, which we show in section 3.3 (now section 4.4) to have a significant effect for some samples. As shown in Fig. 5h and 10a, increasing the number of channels (until reaching ~1800 Gy) does not lead to a $D_e$ plateau and we caution against the assumption that a higher number of channels will necessarily be optimal for dating.

- Figure 9: It requires an illustration of the separated dose signal components. Currently, it appears a little bit arbitrary and descriptive.

➔ We have added a legend showing the different signal segments.

- Figure S4: The offsets of the curves are a little bit difficult to see, perhaps you can show the residuals?

➔ The residuals were added.

- Figure S9: What does this figure add to the manuscript? Your concern is the comparison of two methods, here you compare all kind of protocols and procedures on top of two types of IR-RF and IRPL. I can somewhat understand your Fig. 6 in the main text, but this seems too much.

➔ This figure only compares how results from two protocols plot against the expected values: in panel (a) the IRPL protocol (which contains sequential IRSL and IRPL steps) and (b) the IR-RF protocol. We have added a sentence to the caption to clarify this. Our IRPL measurements were undertaken to increase the dataset presented by Kumar et al. (2021), so we used their MET pIRIR-IRPL protocol. Since our study's focus is the comparison of IR-RF and IRPL, the IRSL $D_e$ are not discussed in the main text, but we show them in the supplement so as not to ignore the data.

**Tables**

- Table 1: Instead of 'se' that refers to the standard error (of what?), please use confidence intervals.

➔ The expected ages and their associated uncertainties (standard error of the mean) are taken directly from the papers in which they were dated, so we have kept them to remain comparable with the previous work.

- Table 3: What do the uncertainties represent? The standard deviation? For such a low 'n' you should rather calculate confidence intervals using the *t*-distribution unless you can show that the normal distribution approximation is valid.

➔ The uncertainties of the $D_e$ estimated in this work are the standard error of the mean, which has been added to the caption. Whereas we agree that the t-distribution confidence intervals would be statistically more accurate, we chose to retain the standard error to stay comparable with the studies against which we compare our results, i.e., Buylaert et al. (2009) and Kumar et al (2021). We have calculated the confidence intervals for the main data set and confirm that the same samples match the expected ages, so our conclusions are not affected by the choice of distribution. We have also added the confidence intervals to the new KDE plots in the Supplement. Since the number of aliquots are given for each estimated $D_e$, interested readers have all the parameters to calculate confidence intervals for the other data sets.

- Table 4: Please explain the meaning of the uncertainties and align them. I suggest calculating consistently 95% confidence intervals.

➔ We have added to the caption that the uncertainties represent the standard error of the mean. As for the suggested confidence intervals, see our response above.

**References**

Autzen, M., Andersen, C.E., Bailey, M., Murray, A.S., 2022. Calibration quartz: An update on dose calculations for luminescence dating. Radiation Measurements 106828. [https://doi.org/10.1016/j.radmeas.2022.106828](https://doi.org/10.1016/j.radmeas.2022.106828)

Buylaert, J.P., Jain, M., Murray, A.S., Thomsen, K.J., Lapp, T., 2012. IR-RF dating of sand-sized K-feldspar extracts: A test of accuracy. Radiation Measurements 47, 759–765. [https://doi.org/10.1016/j.radmeas.2012.06.021](https://doi.org/10.1016/j.radmeas.2012.06.021)

Duller, G.A.T., 1992. Luminescence chronology of raised marine terraces, south-west north island, New Zealand (PhD thesis). University of Wales, Aberystwyth.

Duval, M., Guilarte, V., a, I.C. n, Arnold, L.J., Miguens, L., Iglesias, J., lez-Sierra, S.G. a, 2018. Quantifying hydrofluoric acid etching of quartz and feldspar coarse grains based on

weight loss estimates: implication for ESR and luminescence dating studies. Ancient TL 36, 1–14. [https://doi.org/10.26034/la.atl.2018.522](https://doi.org/10.26034/la.atl.2018.522)

Erfurt, G., Krbetschek, M.R., 2003. IRSAR - A single-aliquot regenerative-dose dating protocol applied to the infrared radiofluorescence (IR-RF) of coarse-grain K-feldspar. Ancient TL 21, 35–42. [https://doi.org/10.26034/la.atl.2003.358](https://doi.org/10.26034/la.atl.2003.358)

Frouin, M., Huot, S., Kreutzer, S., Lahaye, C., Lamothe, M., Philippe, A., Mercier, N., 2017. An improved radiofluorescence single-aliquot regenerative dose protocol for K-feldspars. Quaternary Geochronology 38, 13–24. [https://doi.org/10.1016/j.quageo.2016.11.004](https://doi.org/10.1016/j.quageo.2016.11.004)

Guralnik, B., Jain, M., Herman, F., Ankjærgaard, C., Murray, A.S., Valla, P.G., Preusser, F., King, G.E., Chen, R., Lowick, S.E., Kook, M., Rhodes, E.J., 2015. OSL-thermochronometry of feldspar from the KTB borehole, Germany. Earth and Planetary Science Letters 423, 232–243. [https://doi.org/10.1016/j.epsl.2015.04.032](https://doi.org/10.1016/j.epsl.2015.04.032)

Krbetschek, M.R., Trautmann, T., Dietrich, A., Stolz, W., 2000. Radioluminescence dating of sediments: methodological aspects. Radiation Measurements 32, 493–498. [https://doi.org/10.1016/s1350-4487(00)00122-0](https://doi.org/10.1016/s1350-4487(00)00122-0)

Kreutzer, S., Murari, M.K., Frouin, M., Fuchs, M., Mercier, N., 2017. Always remain suspicious: a case study on tracking down a technical artefact while measuring IR-RF. Ancient TL 35, 20–30. https://doi.org/10.26034/la.atl.2017.510

Lapp, T., Jain, M., Thomsen, K.J., Murray, A.S., Buylaert, J.P., 2012. New luminescence measurement facilities in retrospective dosimetry. Radiation Measurements 47, 803–808. [https://doi.org/10.1016/j.radmeas.2012.02.006](https://doi.org/10.1016/j.radmeas.2012.02.006)

Murari, M.K., Kreutzer, S., Fuchs, M., 2018. Further investigations on IR-RF: Dose recovery and correction. Radiation Measurements 120, 110–119. [https://doi.org/10.1016/j.radmeas.2018.04.017](https://doi.org/10.1016/j.radmeas.2018.04.017)

Murari, M.K., Kreutzer, S., Frouin, M., Friedrich, J., Lauer, T., Klasen, N., Schmidt, C., Tsukamoto, S., Richter, D., Mercier, N., Fuchs, M., 2021. Infrared Radiofluorescence (IR-RF) of K-Feldspar: An Interlaboratory Comparison. Geochronometria 48, 95–110. [https://doi.org/10.2478/geochr-2021-0007](https://doi.org/10.2478/geochr-2021-0007)

Peng, J., Wang, X., Adamiec, G., 2022. The build-up of the laboratory-generated dose-response curve and underestimation of equivalent dose for quartz OSL in the high dose region: A critical modelling study. Quaternary Geochronology 67, 101231. [https://doi.org/10.1016/j.quageo.2021.101231](https://doi.org/10.1016/j.quageo.2021.101231)

*Prescott, J.R., Huntley, D.J., Hutton, J.T., 1993. Estimation of equivalent dose in thermoluminescence dating - the Australian slide method. Ancient TL 11, 1–5. [https://doi.org/10.26034/la.atl.1993.204](https://doi.org/10.26034/la.atl.1993.204)*

*Porat, N., Faerstein, G., Medialdea, A., Murray, A.S., 2015. Re-examination of common extraction and purification methods of quartz and feldspar for luminescence dating. Ancient TL 33, 22–30. [https://doi.org/10.26034/la.atl.2015.487](https://doi.org/10.26034/la.atl.2015.487)*
* * *
**RC2**

➔ We thank the reviewer for their positive assessment of our work. We have responded to the raised issues below, where the original comments are in grey font and the responses in black prefaced by an arrow.

Summary

Infrared radiofluorescence (IR-RF) and infrared photo luminescence (IRPL) are two relatively new innovative luminescence dating techniques that exploit the principal trap. They have the potential to circumvent the disadvantages of traditional infrared stimulated luminescence as they do not rely on the mechanism of recombination and thus have been observed to not suffer from significant fading (Kumar et al., 2022). Since the introduction of IRPL there have been questions as to how these remarkably similar signals compare with regards to dating utility and accuracy. This manuscript is a well written presentation of a series of experiments that initially include recent methodological improvements of the IR-RF signal (elevated temperature measurements and the correction of sensitivity change using vertical sliding), as well as narrowing the bandpass filter and introducing a multiple-aliquot regenerative (MAR) protocol approach; and there after compares these improvements in the IR-RF results with the previously reported IRPL results from Kumar et al. (2021) and new IRPL measurements. This manuscript addresses many important, long-speculated questions about the IR-RF signal in a systematic and thorough manner.

This manuscript fully aligns with the scope of Geochronology and there is no question that it should be published. I have a few minor comments/questions and general remarks that I am confident will be easily addressed and improve the manuscript.

General remarks

- The introduction is easy to read and well written. However, in the last paragraph it could benefit from a clearer set of hypothesis statements that can be tested and evaluated. While there is an implied expectation that the re-analysis of previous data, remeasurement of previously measured samples, the narrowing of the emission and the introduction of the MAR protocol are all intended to improve the previously published IR-RF results for this data set; this could be more explicit.
    ➔ We have changed the final paragraph, as suggested.

- Large swaths of the manuscript feel a bit cobbled together or tacked on. Namely: the bleaching tests, single grains, and sliding range sections. The bleaching test section could be shortened considerably or even incorporated into the $D_e$ determination section. The single grains section is very interesting but it does seem to appear without warning and then isn't mentioned later. As the single grains section and the sliding range section are both linked with the initial rise aspect (a contribution to the cause of the initial rise of the signal and avoiding this part of the curve by shifting and removing the initial channels) with a little reorganization they could potentially be incorporated in the same section.

  ➔ We have restructured the results section, bringing the bleaching test forward to be a more direct comparison with the bleaching test of Buylaert et al. (2012). We have also added a new section that now includes the single grain results and a short literature review, which informs on the segments removed in the sliding range section to avoid the initial rise.

- The section on the narrowing and shifting of the bandpass filter while interesting and a necessary line of enquiry could be more succinct, it draws focus away from the main thread of the paper as the change does not significantly improve the De values.

  ➔ We have shortened the description of results in this section since there was indeed no improvement on $D_e$ accuracy.

- The MAR protocol section is an interesting and innovative approach that could potentially be extended to an entirely separate paper without taking anything away from this manuscript.

  ➔ We have chosen to retain the section here to serve as a motivation for future research, as we believe these preliminary results already suggest the method's potential. We currently do not have the research capacity to pursue this direction into its own study.

- The use of well documented and known samples is an advantage that adds a richness and gravity to this manuscript that is appreciated.

  ➔ Thank you!

Minor comments:

L 20: Our results, following the IR-RF70 protocol with sensitivity corrections, show…

  ➔ Corrected.

L33: the 'NB' in parenthesis is not necessary

  ➔ 'NB' removed.

L88: '…IR-RF De values/results obtained…'

➔ Added 'values'.

 Please refer the reader to table 1, or alternatively include a figure simplifying which samples are used for which measurements and which publications they were previously used in. Such a figure would be very useful later in the paper as well.

➔ We have added references to Table 1 and to a new figure in the supplement, which summarizes papers containing other relevant results using the same samples.

L102: K-feldspar samples are not traditionally etched with HF. Since you measured previously prepared samples, and clearly should be directly comparing the samples, this is unavoidable. Perhaps a comment on this is necessary.

➔ We have added a paragraph to section 2.1 discussing that we do not expect the different treatments to have affected our conclusions, since all samples expected to be in the dating range (i.e., not saturated) received the same HF treatment.

L112: What was the size of these newly measured large aliquots, 6mm? Is this consistent with the previously measured aliquots?

➔ We have added estimates of the aliquot sizes to the main text. The new measurements used aliquots of ca. 3–4 mm in diameter, which we believe to be comparable to the large aliquots of ~8 mm used by Buylaert et al. (2012) because in both cases the signal would be averaged over at least several dozens of grains, even if only a small proportion emits a signal.

L118: It is appreciated that the calibration quartz batch number is reported for the Gießen re-measurements, was the same calibration quartz used across all measurements? And the previously published measurements? Autzen et al. (2022) has reported that different batches of Risø calibration quartz supplied over the years have not been entirely accurate and require a correction of up to 8.25 %. It may be worth discussing this and adding to the uncertainty of the samples where needed.

➔ The same calibration was used for all measurements in Gießen, we have added the dose rate to section 2.2. For the re-analysed measurements from Buylaert et al. (2012) the dose rate used for each aliquot is given in the associated data (Sontag-González et al., 2024). The same calibration quartz type was used, but an earlier batch. We have now corrected them using the 8.25% value suggested by Autzen et al. (2022). We also used this value to correct the control ages that had been estimated using OSL.

L124: aliquot size?

➔ Added (~4 mm); see also comment for line 112.

Fig. 1: A purely aesthetic suggestion, the grey bars illustrating the shift are a bit washed-out a darker/brighter colour might be preferable. Additionally, an indication on the x-axis of the $D_e$ could be helpful to the reader.

➔ We have changed the colour scheme and added an arrow indicating the $D_e$ value on the x-axis.

L163: What is the reason for the double exponential functions? Is this consistent with the previously published IRPL measurements? Or is it just a better statistical fit?

➔ We chose a double exponential function to be able to directly compare our results with those of Kumar et al. (2021). We now state this in section 2.4.

L207-214: It could be helpful to the reader to include the number of channels beside the dose, at least in a few places. In every instance would be overwhelming.

➔ We have added the number of channels in key places.

L245: And so ˜600 Gy (1000 channels)" is the recommended natural measurement length?

➔ After some re-considerations, we have changed the values in Table 4 to show the $D_e$ obtained from a ~300 Gy (500 channels) segment. The 300 Gy and 600 Gy segments yield very similar results, but the 300 Gy segment lies at the very beginning of the observed $D_e$-increase, so at this time we are recommending this segment. However, until we understand what causes the rise in $D_e$ with increasing segment length, we cannot say what the right segment is.

Figure 3: Though busy, is well structured with good use of colour. It went a long way to clarifying the text.

➔ Thank you.

L284: '…differences between the two filters…'

➔ Change accepted.

L437:' We also recommend removing up to ~35 Gy of the initial signal…' this word order is clearer. Perhaps emphasize that this should be aliquot specific.

➔ We have rephrased the sentence.

Tables: Please clarify how the standard errors/standard deviations were calculated.

➔ The uncertainties of the $D_e$ estimated in this work are the standard error of the mean, which has been added to the table caption.

L486: The natural DRC from Buchanan et al., (2022) of the Chinese Loess sequence was built based on several different samples making up the different data points (each sample an average of a series of aliquots) from the tested sequence and though normalised, is unlikely to be directly comparable to the DRC of a single sample.

➔ We agree this is a very rough comparison. However, in Fig. 8, we show that the IR-RF DRC shape is very reproducible across samples (with two exceptions in ten samples), which supports the comparability of DRCs.

**References**

Autzen, M., Andersen, C.E., Bailey, M., Murray, A.S., 2022. [Ca]libration quartz: an update on dose calculations for luminescence dating. Rad. Meas., 157. https://doi.org/10.1016/j.radmeas.2022.106828

Buchanan, G. R., Tsukamoto, S., Zhang, J., and Long, H., 2022. Testing the natural limits of infrared radiofluorescence dating of the Luochuan loess-palaeosol sequence, Chinese Loess Plateau, Radiat. Meas., 155, 106797. https://doi.org/10.1016/j.radmeas.2022.106797, 2022.

Kumar, R., Kook, M., and Jain, M., 2021. Sediment dating using Infrared Photoluminescence, Quat. Geochronol., 62, 101147,570. https://doi.org/10.1016/j.quageo.2020.101147

Kumar, R., Kook, M., Jain, M., 2022. Does hole instability cause anomalous fading of luminescence in feldspar? J. of Lum, 252. https://doi.org/10.1016/j.lumin.2022.119403

Sontag-González, M., Murari, M. K., Jain, M., Frouin, M., and Fuchs, M.: Further investigations into the accuracy of infrared-radiofluorescence (IR-RF) and its inter-comparison with infrared photoluminescence (IRPL) dating (v1.0.1) [Data set]. Zenodo. https://doi.org/10.5281/zenodo.14507179, 2024.

---

## Author Response (AR2)

We thank the reviewers and editors for their feedback. In addition to their comments, we made the following changes:

- Corrected Fig. S11d, which had the wrong dataset plotted for segment 2–75 Gy.
- Corrected the files in the associated supplemental data for sample Gi326.

**Associate editor**

Section 3.1 and 3.2: Initial signal rise
This gets me back to the old question we were asking, why it never not happened in the measurements of Krbetschek's lab. Do you think it is due to their preperation technique (flotation)?

Reply: We did a preliminary test on a sample that had undergone flotation (not yet published) and also observed the initial rise, so it is probably not an issue of sample preparation only. Currently, our hypothesis is that the difference was caused by the detection systems (filter+PMT vs. spectrometer), though we do not have the data to prove this. It is our understanding that the expected transmission of filters is based on a normal incidence (i.e., 90° to the filter surface) and that other angles shift the transmission to shorter wavelengths (e.g., https://www.edmundoptics.com/knowledge-center/application-notes/optics/optical-filter-orientation/?srsltid=AfmBOorEUmySJtiMhXOV3G9vc6xriml0lJ3Y1MZ9li48DBzTxPlFnaO8). So, there may be some stray transmission of highly angled emissions outside the desired detection window. Using a spectroscopic detection, this issue would be suppressed. Below, we have added a very rough comparison of both detection systems for the same samples from previously published data. It is by no means definitive, but the initial rise seems to be somewhat reduced in the spectrometer system (note that the channel lengths were twice as long for the spectrometer measurement).

[Figure]

Fig. 1: Regenerative dose response curves obtained using (a) a PMT and an 850/40 nm filter or (b) a spectrometer, modelling the transmitted signal according to the filter and PMT transmission datasheets. RF is normalised to the highest signal intensity. Different aliquots were used for each panel. The underlying data are shown in Sontag-González and Fuchs (2022) (a) Fig. 1c and (b) Fig. S1 g–h.

Section 5 MAR
Even after following the discussion between the authors and reviewers, the MAR results seem to be

most promissing. And you clearly understand the logic behind it. Personally I would use the MAR data to be the representative of IR-RF in Fig. 6b and Fig. S13. (but feel free to disagree!)

Reply: Thanks! It's very encouraging to hear that. We have added the MAR data to Fig. S13 for completion but take a more conservative approach to Fig. 6b, as we feel the method is still too preliminary to be considered the most reliable of the IR-RF methods. In particular, we still see a change in $D_e$ with progressively longer segments, indicating that the issue is not solved. For better comparability, we have also added a new figure to the supplement (Fig. S15) with a summary of the change in MAR $D_e$ with segments equivalent to Fig. 10 for the SAR results.

Tables 3 and 4
Please use a consistent significant number of digits. Especially the inconsistency between the IR-RF and IRPL data do not look good (up to 4 digits in IR-RF and up to 3 digits in IRPL except Gi326).

Reply: We reduced the number of significant digits to match the IRPL data (taken directly from Kumar et al. (2021)).

**Anonymous referee #1**

**# General remarks**

The authors have substantially revised their manuscript and made significant efforts to address all suggestions. The findings are supported by the data and the manuscript follows established standards for data transparency and reporting.

Therefore, I recommend the publication of the manuscript in Geochronology, pending the incorporation of a few comments I have listed below (naturally do authors do not have to agree to all of them). Although I have now flagged some minor issues I had overlooked the last time, I do not need to see the manuscript again.

**# General comments**

* I had overlooked that last time that the authors have reported expected ages in Table 1, but never reported the comparison (IR-RF/IRPL vs expected) ages. Please add a table or plot for this; as for the approach, the authors can pick the one they prefer; however, it should be consistent. If it does not fit the main text, please report it in the supplement and refer to it. I had stumbled over it because the authors mention in their abstract and age span but then only work with equivalent doses, while the readers would be most likely more interested in the final age comparison.
Reply: Our findings are expected to relate only to the equivalent dose (i.e., irrespective of sample dose rate), so we have added the expected doses to Table 1 and rephrased the sentence in the abstract to highlight the sample doses rather than the ages: "For four out of the seven tested known-age samples spanning ca. 100–300 Gy (20–130 ka), we obtained results in keeping with the expected doses. Two additional modern samples, however, yielded slight dose underestimations."

**# Response to author's responses**

> We have commented on this issue in section 2.1, but since all samples expected to be
> in the dating range received the same treatment, we do not expect etching to affect our
> conclusions.

Perhaps you can rephrase the 2nd part of your addition in Sec. 2.1? I find it confusing that you connect the 'dating range' with the 'same treatment' to conclude that your results are not affected. I think the reader can guess what you mean, but it is certainly not straightforward to understand.

Reply: We have rephrased for clarity: "The effect on the resulting $D_e$ is poorly studied, but all samples expected to be in the dating range (i.e., not saturated) received the same treatment (HF etching), so any variation in $D_e$ accuracy we observe for these samples would not be caused by a difference in sample preparation. Thus, we do not expect etching to affect our conclusions."

> In any case, we have rephrased this sentence to "a more athermally stable signal".

The information provided deviates from the content of Krbetschek et al. (2000) and should not be attributed to the authors. Please adhere to the original text or provide a different reference.

Reply: We have rephrased to: "The main advantages of IR-RF dating over the more common infrared stimulated luminescence (IRSL) of K-feldspar (Hütt et al., 1988) include a more athermally stable signal (based on IR-RF fading tests by Krbetschek et al., (2000) suggesting signal stability)…" according to the statement by Krbetshek et al. (2000): "Fading tests (storage over periods of several months at room temperature) have shown signal stability" (p. 497).

> We did not state the number of rejected channels here in the methods section because
> this was a parameter we varied. We have added a sentence stating that between 0 and
> 499 channels were ignored, which will be [...]

Those figures and experiments you have produced are excellent, and I suggest that you add them to your supplementary material (except for the one you have in the main text anyway) because readers will unlikely look up all the discussion.

Reply: Thank you! We have added the dose vs. signal figure to the supplement.

**Detailed comments**

**My line numbers refer to the version with the changes tracked.**

* L24: I think that, given the current understanding of IR-RF, it is also trap specific and the authors seem to confirm this multiple times in the manuscript. Please rephrase.

Reply: We have rephrased and added a separate sentence: "Like IR-RF, IRPL is also expected to be trap-specific."

* L22-L24: Given the IRPL appears in the title of your manuscript and is an essential part, the phrase feels oddly formulated. Please prioritise it according to your manuscript content.

Reply: Our results in the manuscript are more heavily focussed on IR-RF than IRPL dating, so we feel that the abstract is correctly proportioned between the two methods. Unfortunately, this proportion is more difficult to portray in the title.

* L39: Add proper reference for the SAR approach and more correctly you should refer to the IRSAR approach (Erfurt and Krbetschek, 2003) where applicable as this approach does not use a test dose for sensitivity correction, while it is an essential parameter in the original SAR approach by Murray

and Wintle (2000)

Reply: True, that is an important distinction. We have added the reference: "(IRSAR; Erfurt and Krbetschek, 2003)

* L89: I had somewhat overlooked this the last time. What makes you believe that 'signal instability' is the primary reason for the observed saturation? Out of the many possibilities, this is the one with the lowest explanatory power, given that we can indeed successfully date events using IR-RF and IRPL. If the principle trap is indeed unstable, this shouldn't be possible.

Reply: We have removed the suggestion.

* L125: Subtle issue: Autzen et al. (2022) reported the corrections, Tribolo et al. (2019) flagged the 'issue'.
Reply: We have changed the reference.

* L129: Minor wording inconsistency: The factor you applied reads 1.0825.
Reply: Thanks for catching that! Corrected.

* L135: Also here, 8.25\% can mean reduction or a boost.
Reply: Corrected, as above.

* L211: Why over four million? The formula should be n(n-1)/2 -> 1500*1499/2 -> 1,124,250 (ordered permutations for a segment length of 2 channels). Besides, I suggest removing the sentences starting from 'Since [...]' they read verbose and add little to manuscript.

Reply: We have removed the sentence, as suggested, but we note that we had 3000 channels (so, 3000*2999/2=4498500).

* L226: I suggest removing 'These results [...]' because it is indeed a suggestive statement that is certainly true for all multi-grain luminescence measurements but it depends on many factors unrelated to the 'multi-grain' nature.
Reply: It is true that variation at the single-grain level can lead to issues at the multi-grain level for all multi-grain luminescence measurements, so we have changed the sentence to specify that we mean only IR-RF measurements. However, we have chosen to retain the sentence, as it introduces the modelling in the rest of the paragraph.

* L230: I propose removing this sentence, as it could potentially be misinterpreted and used as a justification odd data treatments. By removing it, you can avoid the risk of it being misinterpreted as a cure for the symptoms rather than using remedy, for instance, better sample preparation and potentially signal deconvolution. If you wish to retain the phrase, it is essential to clearly state the problems associated with it. Specifically, any removal of initial parts of the curve is somewhat arbitrary and should not be used to "tune" the results.
Reply: We have retained the observation because it is the basis for some investigations in later segments but we have rephrased the conclusions to avoid misinterpretation: "This suggests that (i) the 'initial signal rise' originates from signal contamination by presumably non-K-feldspar minerals and (ii) the DRC of the modelled 'contaminated' aliquot converges with that of the 'pure' one (for this sample at ~100 Gy)."

* L595 onwards: Please repeat in one sentence the mission of your work before jumping the summary. Furthermore, to help the reader grasping all your results, you should bullet point your investigations and then conclude the outcome in one, maximum two sentences per point.
Reply: We have added a short summary of our intent before the results summary, as suggested: "We tested whether the methodological developments of the past decade have improved the accuracy of IR-RF dating of known age samples which had previously yielded inaccurate IR-RF ages with an IRSAR protocol. Specifically, we re-analysed previous data and re-measured samples using improved measurement and data analysis protocols (i.e., increased measurement temperature and vertical sliding) as well as using new methods (i.e., MAR IR-RF and IRPL)." However, we chose to retain the full-text version of the results summary.

* L659-L661: Journal and DOI missing.
Reply: Added.

* L682-L684: DOI missing.
Reply: Added.

* L691-L693: DOI missing.
Reply: Added URL.

* L732: Canonical URL entry missing
Reply: Added.

**Tables and figures**

* Table 1: Where the age is around zero (055642, 102011) you should write ~0 or <0.1 ka; the quoted negative age (in particular with that precision) does not make sense.
Reply: We have changed it to "ca. 0".

**Comments supplement**

* Figure S2. This plot illustrates that a KDE may not always be meaningful if individual uncertainties are significant (in particular: Gi326). It also demonstrates that the quoted uncertainties in Table 4 are misleading for the given sample. The individual uncertainties are important (except of sample H22553) and should not be disregarded. Regardless of previous studies, it is crucial to provide meaningful uncertainties supported by the data.
Reply: We thank the reviewer for this comment. Indeed, the uncertainty of Gi326 in Fig. S2 did not match that reported in Table 4. We double-checked the data and found we had forgotten to include the 5% source calibration to the uncertainty of this sample (for the other samples it had been included). We have now corrected Table 4 and Figs. 5,7,9 and S9 to include the correct uncertainties. Additionally, we agree that uncertainty-weighted central estimates are an important metric. However, since the focus of this study was the comparison with published values rather than dating, and considering that we did not have the underlying data from the previously published IRPL results, we think the reported uncertainties (after the correction) are sufficient.

* Figure S13: In particular the Abanico plot is rather essential and should not be hidden in the supplement but appear in the main text. If you don't want to remove a main figure, add it to Fig. 6.

Reply: We have added the abanico plot as panel c to Fig. 6, as suggested. Note that this version is slightly different because we noticed we had not used the corrected expected D_e values (source dose rate calibration issue) in the first version.

**References**

Autzen, M., Andersen, C.E., Bailey, M., Murray, A.S., 2022. Calibration quartz: An update on dose calculations for luminescence dating. Radiation Measurements 106828. https://doi.org/10.1016/j.radmeas.2022.106828

Erfurt, G., Krbetschek, M.R., 2003. IRSAR - A single-aliquot regenerative-dose dating protocol applied to the infrared radiofluorescence (IR-RF) of coarse-grain K-feldspar. Ancient TL 21, 35–42. https://doi.org/10.26034/la.atl.2003.358

Murray, A.S., Wintle, A.G., 2000. Luminescence dating of quartz using an improved single-aliquot regenerative-dose protocol. Radiation Measurements 32, 57–73. https://doi.org/10.1016/s1350-4487(99)00253-x

Tribolo, C., Kreutzer, S., Mercier, N., 2019. How reliable are our beta-source calibrations? Ancient TL 37, 1–10. https://doi.org/10.26034/la.atl.2019.529

###

Sontag-González, M. and Fuchs, M.: Spectroscopic investigations of infrared-radiofluorescence (IR-RF) for equivalent dose estimation, Radiation Measurements, 153, 106733, https://doi.org/10.1016/j.radmeas.2022.106733, 2022.